# Free spermidine evokes superoxide radicals that manifest toxicity

**Vineet Kumar[1†‡], Rajesh Kumar Mishra[1†], Debarghya Ghose[1], Arunima Kalita[1], Pulkit Dhiman[1,2], Anand Prakash[1], Nirja Thakur[1], Gopa Mitra[3], Vinod D Chaudhari[1,2], Amit Arora[1*§], Dipak Dutta[1*]**

[1]Division of Molecular Biochemistry and Microbiology, CSIR Institute of Microbial Technology, Chandigarh, India; [2]Division of Medicinal Chemistry, CSIR Institute of Microbial Technology, Chandigarh, India; [3]Clinical Proteomics Unit, Division of Molecular Medicine, St. John's Research Institute, St John's Medical College, Bangalore, India

**\*For correspondence:**
aarora.pgi@gmail.com (AA);
dutta@imtech.res.in (DD)

[†]These authors contributed equally to this work

**Present address:** [‡]Regional Centre for Biotechnology, Faridabad, India; [§]Department of Medical Microbiology, Post Graduate Institute of Medical Education and Research, Chandigarh, India

**Competing interest:** The authors declare that no competing interests exist.

**Abstract** Spermidine and other polyamines alleviate oxidative stress, yet excess spermidine seems toxic to *Escherichia coli* unless it is neutralized by SpeG, an enzyme for the spermidine *N*-acetyl transferase function. Thus, wild-type *E. coli* can tolerate applied exogenous spermidine stress, but Δ*speG* strain of *E. coli* fails to do that. Here, using different reactive oxygen species (ROS) probes and performing electron paramagnetic resonance spectroscopy, we provide evidence that although spermidine mitigates oxidative stress by lowering overall ROS levels, excess of it simultaneously triggers the production of superoxide radicals, thereby causing toxicity in the Δ*speG* strain. Furthermore, performing microarray experiment and other biochemical assays, we show that the spermidine-induced superoxide anions affected redox balance and iron homeostasis. Finally, we demonstrate that while RNA-bound spermidine inhibits iron oxidation, free spermidine interacts and oxidizes the iron to evoke superoxide radicals directly. Therefore, we propose that the spermidine-induced superoxide generation is one of the major causes of spermidine toxicity in *E. coli*.

## Editor's evaluation

The authors argue that a polyamine, spermidine, causes the production of reactive oxygen species (ROS) in *Escherichia coli* by oxidizing Fe2+, but spermidine can also be protective against ROS at lower concentrations when bound to other cellular molecules such as RNA. Thus, spermidine has both protective and antagonistic effects on ROS stress, depending on the cellular concentration.

## Introduction

Polyamines are ubiquitously present in all life forms. They tweak a diverse array of biological processes, for example, nucleic acid and protein metabolism, ion channel functions, cell growth and differentiation, mitochondrial function, autophagy and aging, protection from oxidative damage, actin polymerization, and perhaps many more (*Casero et al., 2018*; *Gawlitta et al., 1981*; *Madeo et al., 2018*; *Michael, 2018*; *Miller-Fleming et al., 2015*; *Oriol-Audit, 1978*; *Pegg, 2016*; *Pohjanpelto et al., 1981*; *Tabor and Tabor, 1984*; *Wallace et al., 2003*). The cationic amine groups of polyamines can avidly bind to the negatively charged molecules, such as RNA, DNA, phospholipids, etc. (*Igarashi and Kashiwagi, 2000*; *Miyamoto et al., 1993*; *Schuber, 1989*; *Tabor and Tabor, 1984*). Polyamines have been demonstrated to protect DNA from reactive oxygen species (ROS) such as singlet oxygen, hydroxyl radical (•OH), or hydrogen peroxide ($H_2O_2$) (*Balasundaram et al., 1993*; *Ha et al., 1998a*; *Ha et al., 1998b*; *Jung and Kim, 2003*; *Khan et al., 1992a*; *Khan et al., 1992b*; *LØVaas, 1996*;

*Pegg, 2018*; *Murray Stewart et al., 2018*). Indeed, knocking out polyamine biosynthesis enzymes from *Escherichia coli* and yeast confers toxicity to oxygen, superoxide anion radical ($O_2^-$), and $H_2O_2$ (*Balasundaram et al., 1993*; *Chattopadhyay et al., 2003*; *Eisenberg et al., 2009*).

Most prokaryotes including *E. coli* synthesize cadaverine, putrescine, and spermidine, while higher eukaryotes additionally synthesize spermine. *E. coli* also acquires spermidine and putrescine from the surrounding medium (*Igarashi and Kashiwagi, 2000*; *Miller-Fleming et al., 2015*). However, polyamine in excess is toxic to the organisms unless polyamine homeostasis in the cell is operated at the levels of export, synthesis, inactivation, and degradation (*Miller-Fleming et al., 2015*). Notably, spermine/spermidine *N*-acetyl transferase (SSAT or SpeG), which inactivates spermidine and spermine, constitutes the most potent polyamine homeostasis component of the cells (*Miller-Fleming et al., 2015*).

A tremendous volume of work has been dedicated to unravel the biological importance of spermidine and its homeostasis mechanisms. It has also been known for long that spermidine (or spermine) in excess is toxic to the organisms and viruses (*Pegg, 2013*). It has been proposed that the excess polyamines may affect protein synthesis by binding to acidic sites in macromolecules, such as nucleic acids, proteins, and membrane, and by displacing magnesium from these sites (*Limsuwun and Jones, 2000*; *Pegg, 2013*). However, a precise molecular detail of spermidine toxicity is not yet understood. In this study, we decipher a molecular mechanism of spermidine toxicity in bacteria. We find the intertwined relationships among spermidine toxicity, iron metabolism, and $O_2^-$ radical production in bacteria.

## Results

### Increased cellular spermidine inhibits overall oxidative stress while apparently evoking less harmful $O_2^-$ production

To determine the working concentrations of exogenous spermidine that sufficiently inhibits the growth of Δ*speG*, but not WT strain, we added various amounts of spermidine in the growth medium. WT cells showed a modest reduction in growth up to 6.4 mM of spermidine concentration (*Figure 1—figure supplement 1*). On the contrary, Δ*speG* strain exhibited a striking decrease in growth when supplemental spermidine level was >3.2 mM (*Figure 1—figure supplement 1*). Therefore, we chose spermidine concentration ≥3.2 mM for our further experiments. We performed HPLC analyses to show whether elevated spermidine level in the Δ*speG* strain caused growth inhibition. Supplementation of 3.2 mM exogenous spermidine in the growth medium increased the intracellular spermidine levels in the Δ*speG* strain, while no significant increase was observed in the WT cells (*Figure 1—figure supplement 1*). The SpeG function apparently converted the excess spermidine to $N^1$- and $N^8$-acetyl-spermidines maintaining the level of spermidine in the WT cells (*Miller-Fleming et al., 2015*). The spermidine synthase-defective (Δ*speE*) strain of *E. coli* also acquired spermidine at a low level from the LB medium (*Figure 1—figure supplement 1*).

It is well documented that polyamine spermidine is an anti-ROS agent (*Balasundaram et al., 1993*; *Chattopadhyay et al., 2003*; *Chattopadhyay et al., 2006*; *Ha et al., 1998a*; *Khan et al., 1992a*; *Khan et al., 1992b*; *Pegg, 2018*; *Murray Stewart et al., 2018*). However, all the in vivo studies in the past have been conducted under polyamine deficient conditions to show ROS production, thereby implicating the anti-ROS function of polyamines. Thus, assessing ROS levels both in spermidine-enriched and spermidine-deficient conditions are missing. To address this, we incubated *E. coli* strains with 2',7'-dichlorodihydrofluorescein diacetate (H2DCFDA) and dihydroethidium (DHE) probes, which generate fluorescent compounds reacting with one-electron-oxidizing species. While H2DCFDA is a generic ROS probe that nonspecifically reacts with many ROS, the DHE is somewhat specific to the $O_2^-$ anions in the system (*Chen et al., 2013*; *Kalyanaraman et al., 2012*). The relative mean fluorescence intensity (MFI) of H2DCFDA was increased about 1.5-fold in the spermidine synthase-defective (Δ*speE*) strain, while no change in MFI was observed in the Δ*speG* strain (*Figure 1A*). However, spermidine treatment significantly decreased the H2DCFDA fluorescence in WT, Δ*speG*, and Δ*speE* strains (*Figure 1A*). Interestingly, despite no apparent increase in the spermidine level in WT cells under spermidine stress (*Figure 1—figure supplement 1*), a significant decrease in the H2DCFDA fluorescence was observed (*Figure 1A*). It is possible that the acetylated products of spermidine might have some role in the declined ROS levels causing decreased H2DCFDA fluorescence in spermidine-fed WT cells. Similarly, the relative MFI of DHE probe was increased significantly (1.5-fold) in Δ*speE* strain

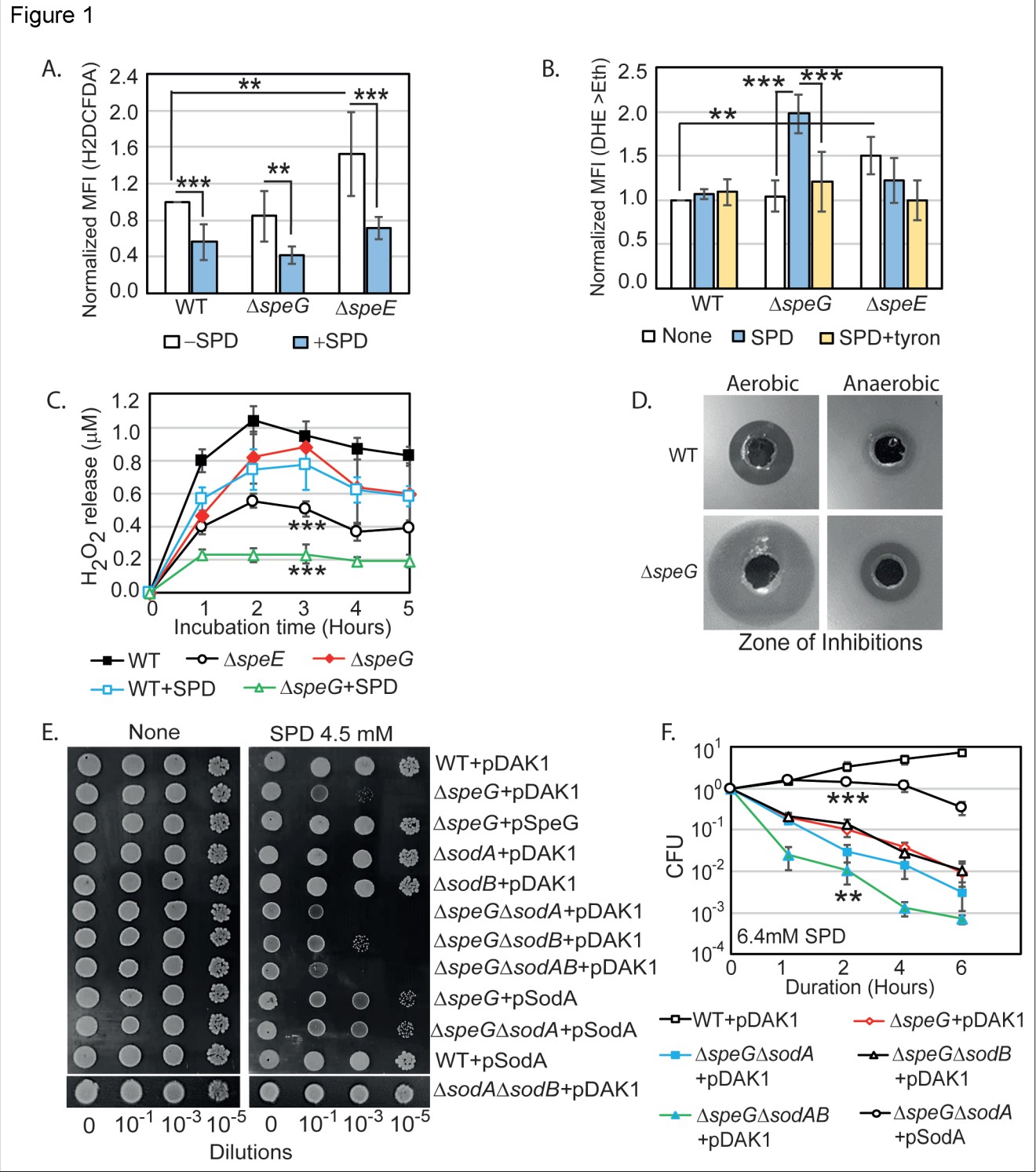

**Figure 1.** Spermidine (SPD) stress and intracellular reactive oxygen species (ROS) in *Escherichia coli.* (**A**) The relative mean fluorescence intensity (MFI) values for the 2',7'-dichlorodihydrofluorescein diacetate (H2DCFDA), which is an indicator of •OH radical production, obtained by flow cytometry analyses are plotted. (**B**) The relative MFI values of dihydroethidium (DHE) probe, which is an indicator of $O_2^-$ radical production, obtained by flow cytometry analyses are plotted. (**C**) The absolute $H_2O_2$ production for a span of 5 hr from the different *E. coli* strains are shown. *** are p-values generated comparing with WT value. (**D**) Zone of inhibitions (ZOIs) surrounding SPD well on the agar plates were shown for the WT and Δ*speG* strains of *E. coli* under aerobic and anaerobic conditions. (**E**) Serially diluted *E. coli* cells were spotted on LB-agar plates to show their sensitivity to SPD. (**F**) Viability of different knockout strains were plotted from the CFU counts in different time intervals after treatment with lethal dose of SPD. ** and

*Figure 1 continued on next page*

*Figure 1 continued*

*** are p-values generated comparing with the values of Δ*speG* and Δ*speG*Δ*sodA*, respectively. Error bars in the panels are mean ± SD from the three independent experiments. Whenever mentioned, *** and ** are <0.001 and <0.01, respectively; unpaired t test. See also *Figure 1—figure supplement 1*, and *Figure 1—source data 1*, *Figure 1—source data 2*, *Figure 1—source data 3*, *Figure 1—source data 4*.

The online version of this article includes the following source data and figure supplement(s) for figure 1:

**Source data 1.** *Figure 1A* Raw data.

**Source data 2.** *Figure 1B* Raw data.

**Source data 3.** *Figure 1C* Raw data.

**Source data 4.** *Figure 1F* Raw data.

**Figure supplement 1.** Spermidine-mediated $O_2^-$ production is apparently toxic in the absence of SpeG function.

**Figure supplement 1—source data 1.** *Figure 1—figure supplement 1C* raw data.

**Figure supplement 1—source data 2.** *Figure 1—figure supplement 1C* raw HPLC peak profiles.

(*Figure 1B*). These findings are consistent with the observations that spermidine is an anti-ROS agent (*Balasundaram et al., 1993*; *Chattopadhyay et al., 2003*; *Chattopadhyay et al., 2006*; *Ha et al., 1998a*; *Khan et al., 1992a*; *Khan et al., 1992b*; *Pegg, 2018*; *Murray Stewart et al., 2018*).

Surprisingly, the relative MFI of DHE probe was increased significantly (2-fold) in the spermidine-fed Δ*speG* as compared to WT strain of *E. coli* (*Figure 1B*). Tyron (Tr), an $O_2^-$ quencher, decreased the MFI of DHE in the spermidine-fed Δ*speG* strain (*Figure 1B*). These observations indicate that although spermidine accumulation in the Δ*speG* strain reduces overall ROS levels and oxidative stress (*Figure 1A*), it may simultaneously evoke less harmful $O_2^-$ production (*Balasundaram et al., 1993*; *Chattopadhyay et al., 2003*; *Chattopadhyay et al., 2006*; *Ha et al., 1998a*; *Khan et al., 1992a*; *Khan et al., 1992b*; *Pegg, 2018*; *Murray Stewart et al., 2018*). In another assay, we determined that the Δ*speE* and spermidine-fed Δ*speG* strains release substantially low levels of $H_2O_2$ compared to the untreated counterpart and WT cells (*Figure 1C*).

Next, we allowed WT and Δ*speG* strains to grow against the spermidine-diffusing wells on agar plates in aerobic and anaerobic conditions (*Figure 1D*). A far wider zone of inhibition (ZOI) of growth for Δ*speG* strain was observed compared to WT under aerobic condition (*Figure 1D*), while a narrow ZOI was observed under anaerobic conditions for both strains (*Figure 1D*). This data further indicates that $O_2^-$ production in aerobic condition could be a cause of the observed spermidine toxicity.

If spermidine induces $O_2^-$ production, superoxide dismutase (SOD) genes (e.g., *sodA* and *sodB*) would play vital roles. Therefore, the serial dilutions of WT, Δ*speG*, Δ*sodA*, Δ*sodB*, and corresponding double and triple mutants, viz. Δ*speG*Δ*sodA*, Δ*speG*Δ*sodB*, Δ*sodA*Δ*sodB*, and Δ*speG*Δ*sodA*Δ*sodB*, were transformed with either empty vector, pDAK1, or pSodA vectors. The Δ*speG*Δ*sodA* and Δ*speG*-Δ*sodA*Δ*sodB* mutants containing empty vector exhibited higher growth defects than Δ*speG* strain on LB-agar plate supplemented with spermidine (*Figure 1E*). However, the cell viability of the double mutants was similar to the Δ*speG* strain, while the triple mutant exhibited an accelerated loss of cell viability, in the presence of spermidine (*Figure 1F*). The multicopy induction of SodA from pSodA plasmid suppressed the growth defect in the Δ*speG* and Δ*speG*Δ*sodA* strains (*Figure 1E*). The over-expression of SodA also improved the viability of Δ*speG*Δ*sodA* strain (*Figure 1F*). Note that, unlike Δ*speG* strain, the single mutants show growth and viability similar to the WT strain in the presence or absence of spermidine (*Figure 1E* and *Figure 1—figure supplement 1*). This data suggests that the absence of SOD enzymes aggravates $O_2^-$ toxicity in the spermidine-fed Δ*speG* strain.

## Spermidine stress evokes $O_2^-$ production in Δ*speG* strain

Although the above experiments apparently suggest for the production of $O_2^-$ anions under spermidine stress, they are not direct and confirmatory in nature, as the ROS probes often reacts with multiple ROS (*Kalyanaraman et al., 2012*). Spermidine transport is a proton motif force (PMF)-dependent process (*Kashiwagi et al., 1986*). Therefore, the observed narrower ZOI in the presence of spermidine under anaerobic condition (*Figure 1D*) could also be due to the low PMF under anaerobic condition. Thus, to determine the relative levels of intracellular $O_2^-$ species, we performed electron paramagnetic resonance (EPR) using a cell-permeable cyclic hydroxylamine spin-probe, 1-hydroxy-3-

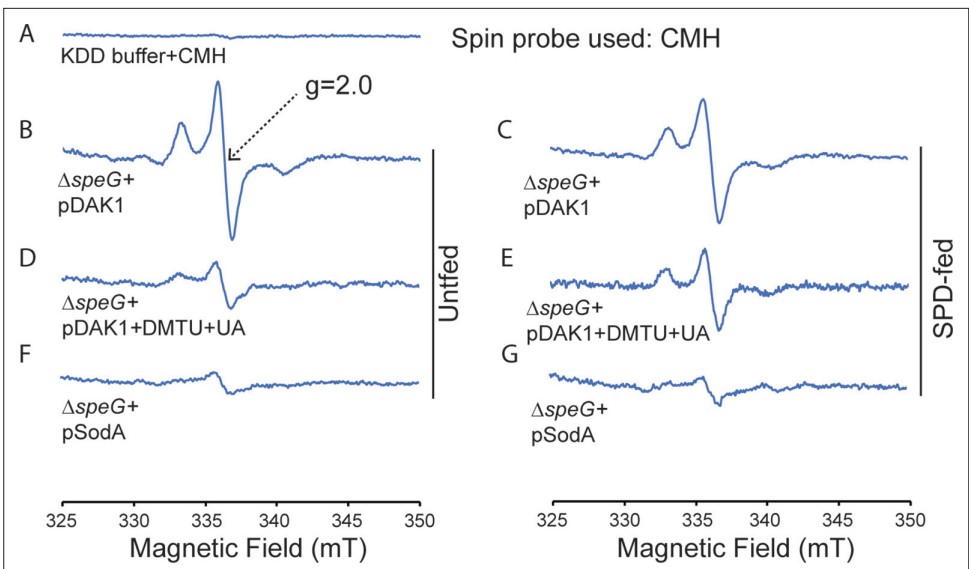

**Figure 2.** Spermidine stress generates $O_2^-$ radical production in $\Delta speG$ strain. (**A**) 1-Hydroxy-3-methoxycarbonyl-2,2,5,5-tetramethylpyrrolidine (CMH) probe incubated with KDD buffer before electron paramagnetic resonance (EPR) analysis. (**B, C, D**) EPR spectra $\Delta speG$ strain with the plasmids pDAK1 (empty vector) or pSodA were grown without spermidine and performed EPR adding CMH probe. (**E, F, G**) $\Delta speG$ strain with the plasmids pDAK1 (empty vector) or pSodA were grown with spermidine and performed EPR adding CMH spin probe, as mentioned in the Materials and methods. See also **Figure 2—source data 1**.

The online version of this article includes the following source data for figure 2:

**Source data 1.** **Figure 2** Raw data.

methoxycarbonyl-2,2,5,5-tetramethylpyrrolidine (CMH) (**Dikalov et al., 2018**). Compared to spin-trap agents, lower level of CMH reacts at a much faster rate with $O_2^-$ anion, producing highly stable and EPR-sensitive nitroxide radicals (**Dikalov et al., 2018**). However, peroxynitrite and •OH radicals can also oxidize CMH (**Dikalov et al., 2018**; **Thomas et al., 2015**).

In the first set of reactions, the unfed and spermidine-fed $\Delta speG$ cells carrying an empty vector were incubated with CMH. In the second set, portions of the unfed and spermidine-fed $\Delta speG$ cells carrying an empty vector were preincubated with dimethyl thiourea (DMTU) and uric acid (UA), the scavengers for the •OH and peroxynitrite ($ONOO^-$) radicals, respectively, before CMH addition. In the third set, the unfed and spermidine-fed $\Delta speG$ cells harboring pSodA plasmid were incubated with CMH. In the first set, a high level of EPR signals were detected with more signal in the unfed sample than the spermidine-fed one (**Figure 2B and C**). This data indicates that the overall ROS production is higher in the absence of exogenous spermidine, which is consistent with the notion that the spermidine is an anti-ROS agent (**Balasundaram et al., 1993**; **Chattopadhyay et al., 2003**; **Chattopadhyay et al., 2006**; **Ha et al., 1998b**; **Khan et al., 1992a**; **Khan et al., 1992b**; **Pegg, 2018**; **Murray Stewart et al., 2018**). In contrast, EPR signal was higher in the spermidine-fed cells than unfed one in the second set (**Figure 2D and E**), suggesting that the signals apparently represent CMH oxidation by $O_2^-$ anions. Finally, the decrease in EPR signals under the multicopy expression of SodA (**Figure 2F and G**) suggests that the signals in the second set were indeed generated from $O_2^-$-mediated oxidation of CMH.

## $O_2^-$ production under spermidine stress affects cellular redox state

Antioxidant chemicals viz. Tr, sodium pyruvate (SP), and thiourea (TU) scavenge $O_2^-$, $H_2O_2$, and •OH, respectively (**Bleeke et al., 2004**; **Franco et al., 2007**). Whereas, *N*-acetyl cysteine (NAC) and ascorbate counterbalance oxidative stress replenishing glutathione levels and donating electrons to reducing partners (**Nimse and Pal, 2015**; **Sun, 2010**). We show that Tr, NAC, and ascorbate, but not

SP and TU, rescued the spermidine-mediated growth inhibition phenotype (*Figure 3A*). This observation further suggests that the $O_2^-$ stress-derived redox imbalance could be the route of spermidine toxicity.

The reduced nicotinamide adenine dinucleotide phosphate (NADPH) is a potent reducing agent. NADPH drives glutathione and thioredoxin cycles, thereby producing reduced forms of glutathione (GST), glutaredoxins, and thioredoxins to cope up with oxidative stress. A large fraction of NADPH in *E. coli* is provided by a glucose-6-phosphate 1-dehydrogenase (Zwf) catalyzed reaction (*Olavarría et al., 2012*). We show that both the growth and viability of Δ*speG*Δ*zwf* double mutant were significantly affected compared to the Δ*speG* strain under spermidine stress (*Figure 3B and C*). Complementing Δ*speG*Δ*zwf* with a plasmid, pBAD-*zwf*, rescues the growth defect and mortality under spermidine stress (*Figure 3B and C*). We compared the levels of the total NADP (NADPt), total glutathione (GSt), and their oxidized (NADP+ and GSSG) and reduced (NADPH and GSH) species in the WT and Δ*speG* strains grown in the absence and presence of spermidine. The relative levels of total and reduced species of NADP and GST were decreased significantly in the spermidine-fed Δ*speG* strain (*Figure 3D and E*). NAD serves as the precursor for NADP production. However, the levels of total (NADt), oxidized (NAD+), and reduced (NADH) did not alter significantly (*Figure 3F*). Nevertheless, the NAD + to NADH ratio was significantly increased in the Δ*speG* strain compared to WT cells (*Figure 3F*). No significant increase of the ratios was observed by adding spermidine in the growth medium of WT and Δ*speG* strain (*Figure 3F*). In consistence with the increased ratio of NAD + to NADH, the level of ATP was declined in Δ*speG* strain compared to the unfed WT (*Figure 3G*). ATP level was further decreased in the spermidine-fed Δ*speG* strain (*Figure 3G*).

## Spermidine blocks the induction of SoxR regulon

To understand the global impact of spermidine toxicity, we performed a microarray experiment on the Δ*speG* strain in the presence and absence of spermidine. The genes that were >2-fold downregulated are involved in flagellar biogenesis, acid resistance, hydrogenase function, nitrogen metabolism, electron transport, aromatic and basic amino acid metabolism, etc. (*Figure 4A* and *Supplementary file 1*). Interestingly, transcription of the genes encoding chaperones, heat shock, and other stress factors (*groL*, *groS*, *dnaK*, *hdeAB*, *ibpAB*, *uspAB*, etc.) was also downregulated under spermidine stress (*Supplementary file 1*). On the other hand, among the highly upregulated category, the genes that encode for the ribosome, RNA polymerase, transcription factors, DNA polymerase, and enzymes for the fatty acid biosynthesis and iron-sulfur cluster (*isc*) biogenesis were prominent (*Supplementary file 1* and *Figure 4A*). These observations indicate that apart from inducing superoxide production (*Figures 1 and 2*), the excess spermidine could interfere with broad cellular processes, such as protein folding and proteostasis, DNA, RNA and lipid metabolisms, and iron-sulfur cluster biogenesis. Many operons regulated by Fis and IHF were activated or repressed in our microarray indicating that spermidine could activate Fis and IHF regulon (*Supplementary file 2*). Performing Fisher's exact test, we show that the differential expression of the Fis-regulated operons was significantly enriched (p-value 0.0023). Corroborating with this finding, we show that Δ*speG*Δ*fis*, but not Δ*speG*Δ*ihfA* strain, generated small colonies upon overnight incubation (*Figure 4—figure supplement 1*), suggesting that the role of Fis regulator is critical under spermidine stress. Quantitative real-time PCR (RT-qPCR) experiment was performed to validate the microarray data partially (*Figure 4—figure supplement 2*).

Iron-sulfur center of SoxR senses the levels of cellular $O_2^-$ or NO (*Fujikawa et al., 2017*; *Hidalgo and Demple, 1994*; *Kobayashi, 2017*; *Liochev and Fridovich, 2011*; *Lo et al., 2012*) and triggers transcription of a set of genes, including *soxS*, *sodA*, and *zwf* (*Touati, 2000*; *Wu and Weiss, 1992*). Surprisingly, none of the three critical genes was found to be activated in the microarray. RT-qPCR analyses verified the unaltered expression of *soxS*, *sodA*, and *zwf* under spermidine stress (*Figure 4—figure supplement 2*). Consistently, using Δ*speG* harboring pUA66_*soxS*, a reporter plasmid expressing *gfpmut2* from the *soxS* promoter ($P_{soxS}$-*gfpmut2*), and RKM1 strain containing a chromosomally fused *lacZ* reporter under *sodA* promoter ($P_{sodA}$-*lacZ*) (*Table 1*), we did not find any transcriptional activation of *soxS* and *sodA* promoters (*Figure 4—figure supplement 2*). Therefore, we suspected whether spermidine in excess blocks the $O_2^-$-mediated activation of SoxR, thereby aggravating $O_2^-$ toxicity. However, an alternative explanation for this observation would be that the redox cycling drugs, but not $O_2^-$, are the efficient activators of SoxR (*Gu and Imlay, 2011*). Therefore, we used menadione, a redox cycling agent and $O_2^-$ generator, to observe the $P_{soxS}$-*gfpmut2* reporter induction and chased it

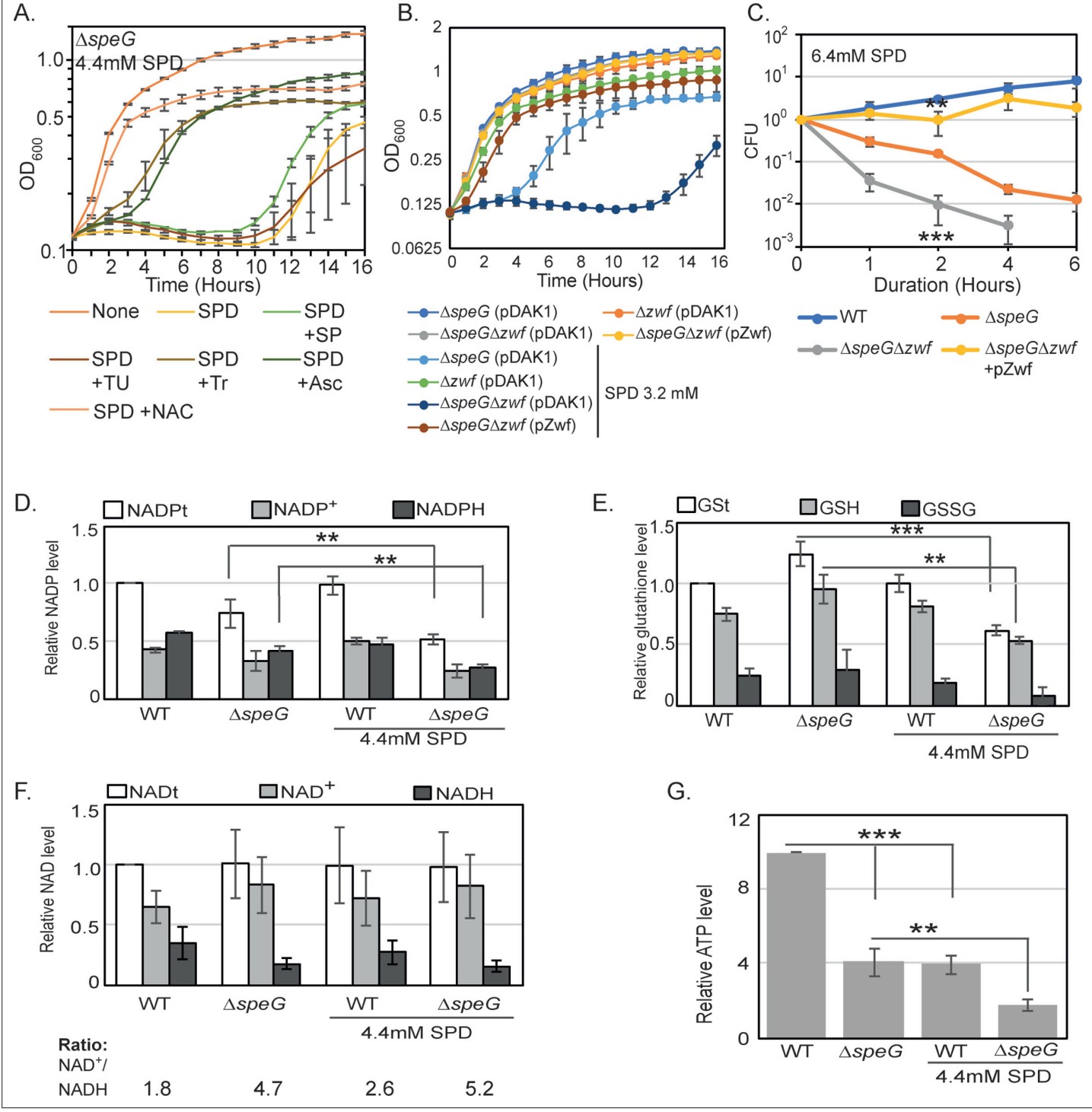

**Figure 3.** $O_2^-$ radical production affects redox balance in the spermidine-fed Δ*speG* strain. (**A**) Growth curves show that Tyron (Tr), ascorbate (Asc), and *N*-acetyl cysteine (NAC) can overcome spermidine (SPD) stress while sodium pyruvate (SP) and thiourea (TU) fail to do so. (**B**) Growth curves show that Δ*speG*Δ*zwf* strain is hypersensitive to SPD in comparison to Δ*speG* strain. Complementation of Δ*speG*Δ*zwf* strain with pZwf plasmid overcomes this SPD hypersensitivity. (**C**) CFUs were obtained for different *Escherichia coli* strains pretreated with SPD for desired time points and plotted to show the reduced viability of Δ*speG*Δ*zwf* strain in comparison to theΔ*speG* strain. (**D**) Relative levels of NADPt and reduced nicotinamide adenine dinucleotide phosphate (NADPH) were significantly decreased in the Δ*speG* strain under SPD stress. (**E**) Relative levels of GSt, GSH, and GSSG were significantly decreased in the SPD-fed Δ*speG* strain. (**F**) No significant change in the relative total NAD (NADt), NAD+, and NADH levels were recorded. However, NAD+ to NADH ratio was significantly increased in the Δ*speG* strain compared to WT cells. No further increase of the ratio was observed by adding SPD in the growth medium of WT and Δ*speG* strain. (**G**) The relative level of ATP was declined in Δ*speG* strain and spermidine-fed WT cells in

*Figure 3 continued on next page*

*Figure 3 continued*

comparison to the unfed WT. SPD supplementation decreased the ATP level further in the SPD-fed Δ*speG* strain. Error bars in the panels are mean ± SD from the three independent experiments. Whenever mentioned, the *** and ** denote p-values < 0.001 and < 0.01, respectively; unpaired t test. See also *Figure 3—source data 2*, *Figure 3—source data 3*, *Figure 3—source data 4*, *Figure 3—source data 5*.

The online version of this article includes the following source data for figure 3:

**Source data 1.** *Figure 3A* Raw data.

**Source data 2.** *Figure 3B* Raw data.

**Source data 3.** *Figure 3C* Raw data.

**Source data 4.** *Figure 3D, E and F* Raw data.

**Source data 5.** *Figure 3G* Raw data.

by spermidine in the Δ*speG* strain. Spermidine also suppressed the menadione-induced GFP reporter fluorescence (*Figure 4B*), suggesting that spermidine indeed blocks SoxR-mediated activation of *soxS* in *E. coli*. A possible mechanism of spermidine-mediated SoxR inactivation is discussed. Among other ROS-responsive genes, the catalase coding genes (*katE* and *katG*) were downregulated (*Figure 4A*), while no change was observed in the expression of *ahpCF* genes under spermidine stress (GEO accession #154618). Using pUA66_*ahpC* and pUA66_*katG* reporter plasmids (P$_{ahpC}$-*gfpmut2* and P$_{katG}$-*gfpmut2*, respectively), we validated these microarray observations (*Figure 4—figure supplement 2*).

Consistent with the microarray expressions, our western blotting experiments exhibited the unchanged expression of SodA and a decreased expression of KatG in the spermidine-treated Δ*speG* strain compared to untreated counterparts (*Figure 4C and D*). However, SodA level was modestly elevated in the Δ*speG* strain, and the spermidine-treated WT strain, in contrast to the untreated WT strain (*Figure 4C*). Contrary to the microarray data, a profound increase in AhpC level was observed while growing WT or Δ*speG* cells in the presence of spermidine, indicating a translational elevation of AhpC level under spermidine stress (*Figure 4E*). Increased AhpC level indicating the activation of alkyl hydroperoxidase (AhpCF) enzyme could be responsible for the decline in cellular $H_2O_2$ level (*Figure 1C*). Thus, declined $H_2O_2$ concentration could be the limiting factor for the cellular •OH radical production under spermidine stress (*Figure 1A*).

## Spermidine affects iron-sulfur cluster biogenesis

$O_2^-$ has the potential to oxidize the solvent-exposed iron-sulfur clusters of *E. coli* dehydratases, aconitase, and fumarase enzymes to liberate free $Fe^{2+}$ (*Benov, 2001*; *Fridovich, 1986*; *Imlay, 2008*). Therefore, supplementation of $Fe^{2+}$ ions helps to repair the damaged clusters (*Gardner and Fridovich, 1992*; *Imlay, 2008*). Consistently, we observed that the declined aconitase activity in the spermidine-stressed Δ*speG* strain was rescued by supplemental $Fe^{2+}$ ion (*Figure 5A*). Besides, the intracellular level of iron in the Δ*speG* strain was decreased more than 3-fold in the presence of spermidine (*Figure 5B*). Consequently, the supplementation of $Fe^{2+}$ salt in the LB-agar plate rescued the growth of spermidine-fed Δ*speG* strain supports this claim (*Figure 5C*).

The iron scarcity was also reflected in the gene expression pattern of IscR regulon (*Figure 4A*). IscR forms a functional holoenzyme with the iron-sulfur cluster. The de-repression of iron-sulfur cluster biogenesis operon (*iscRSUA-hscBA-fdx-iscX*) in the microarray (*Figure 4A*) signifies the presence of non-functional apo-IscR under the scarcity of cellular $Fe^{2+}$ ion (*Esquilin-Lebron et al., 2021*; *Schwartz et al., 2001*). Besides, apo-IscR and apo-Fur activate and derepress the alternative iron-sulfur cluster assembly system (*sufABCDSE*), respectively (*Esquilin-Lebron et al., 2021*; *Outten et al., 2004*). Interestingly, no genes of the *suf* operon were found to be upregulated under spermidine stress. Instead, a 3-fold downregulation of *sufA* was observed (*Figure 4A*). Since *suf* operon is also positively regulated by OxyR (*Esquilin-Lebron et al., 2021*) but spermidine stress declined the cellular $H_2O_2$ levels (*Figure 1C*), we suggest that the combined action of apo-IscR, apo-Fur, and inactivated form of OxyR kept *suf* operon expression indifferent under spermidine stress. Spermidine also activated *rsxA* and *rsxB* (*Figure 4A*), which encode the critical components of the iron-sulfur cluster reducing system of SoxR (*Koo et al., 2003*).

The level of manganese, an antioxidant metal that determines *sodA* activity, is usually increased under iron scarcity (*Kaur et al., 2017*; *Kaur et al., 2014*; *Martin et al., 2015*; *Waters et al., 2011*). However, a modest decrease in the level of cellular manganese under spermidine stress was observed

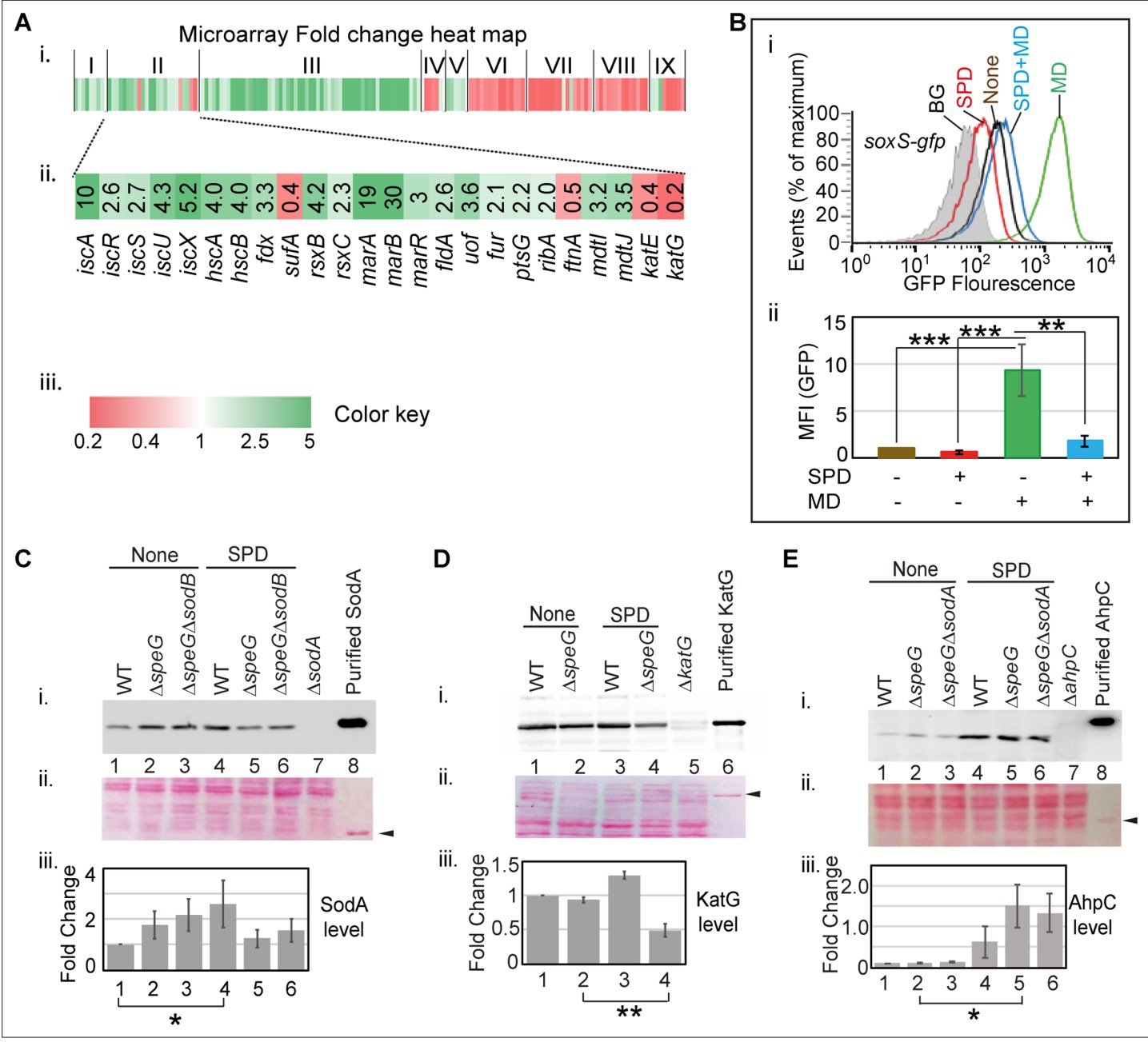

**Figure 4.** Spermidine blocks the activation of superoxide defense circuit. (**A**) (**i**) Microarray heat map showing various categories of genes (I: Replication and transcription associated genes, II: Iron homeostasis, ROS, multidrug resistance and sugar metabolism genes, III: Ribosomal and ribosome biogenesis-associated genes, IV: Oxidoreductase and ATP synthesis genes, V: Fatty acid metabolism-related genes, VI: Flagellar biogenesis-related genes, VII: Acid resistance and chaperone genes, VIII: Hydrogenase and nitrogen metabolizing genes, IX: Amino acid metabolizing genes; see **Supplementary file 1**) that were differentially expressed under spermidine stress. (ii) Zoomed in heat map of the category II genes responsible for iron metabolism and reactive oxygen species (ROS) regulation. (iii) Color key represents the expression fold-change (FC) of the genes. (**B**) The subpanel (**i**) represents a flow cytometry experiment to demonstrate that spermidine (SPD) stress inhibits menadione (MD)-induced P$_{soxS}$-*gfpmut2* reporter fluorescence. The subpanel (ii) represents relative mean fluorescence intensities (MFIs) in the presence or absence of SPD and MD calculated from three different flow cytometry experiments. (C, D, E) Western blotting experiments show SodA, KatG, and AhpC levels in the various strains in the presence or absence of SPD: (**i**) developed blot, (ii) ponceau S-stained counterpart of the same blot, (iii) the bar diagrams represent relative FC of the proteins under SPD stress. The relative FC values were calculated from the band intensity values obtained from three independent blots in comparison to the untreated WT counterparts. Purified 6X His-tagged SodA, KatG, and AhpC proteins were loaded as positive controls. The cellular protein extracts from Δ*sodA*, Δ*katG*, and Δ*ahpC* strains were used for negative controls. Whenever mentioned, the *** and ** denote p-values < 0.001, < 0.01, respectively; unpaired t test. *Figure 4—source data 2.*, *Figure 4—source data 3*, *Figure 4—source data 4*, *Figure 4—source data 5*, *Figure 4—source data 6*, *Figure 4—*

*Figure 4 continued on next page*

*Figure 4 continued*

source data 7, *Figure 4—source data 8*, *Figure 4—source data 9*, *Figure 4—source data 10*, *Figure 4—source data 11*, *Figure 4—source data 12*, *Figure 4—source data 13* and *Figure 4—source data 14*.

The online version of this article includes the following source data and figure supplement(s) for figure 4:

**Source data 1.** *Figure 4B–ii* Raw data.

**Source data 2.** *Figure 4C–i* Raw unedited image.

**Source data 3.** *Figure 4C–i* Raw uncropped and labeled image.

**Source data 4.** *Figure 4C–ii* Raw full image.

**Source data 5.** *Figure 4C–ii* Raw uncropped and labeled image.

**Source data 6.** *Figure 4D–i* Raw full image.

**Source data 7.** *Figure 4D–i* Raw uncropped and labeled image.

**Source data 8.** *Figure 4D–ii* Raw full image.

**Source data 9.** *Figure 4D–ii* Raw uncropped and labeled image.

**Source data 10.** *Figure 4E–i* Raw full image.

**Source data 11.** *Figure 4E–i* Raw uncropped and labeled image.

**Source data 12.** *Figure 4E–ii* Raw full image.

**Source data 13.** *Figure 4E–ii* Raw uncropped and labeled image.

**Source data 14.** *Figure 4C, D and E* Fold change values of the western blots.

**Figure supplement 1.** Indication of the importance of Fis regulator under spermidine stress.

**Figure supplement 2.** Validation of microarray data.

(*Figure 5D*). The low level of manganese could slow down the rate of dismutation of $O_2^-$ anion compromising SodA function, thereby elevating the $O_2^-$ anion levels in the spermidine-treated cells. Finally, we spotted the serially diluted cultures of *E. coli* strains to show that the deletion of two individual genes (*iscU* and *ygfZ*), which are involved in the iron-sulfur cluster biogenesis (*Waller et al., 2010*), affects the growth of the spermidine-treated Δ*speG* strain (*Figure 5E*). Interestingly, the Δ*speG*Δ*soxS* strain was more sensitive to spermidine than the Δ*speG* strain (*Figure 5E*), indicating that the basal level of *soxS* expression has some potential to ameliorate $O_2^-$ under spermidine stress. Although *marA* and *marB* genes were expressed at the highest level in the spermidine-stressed Δ*speG* strain (*Figure 4A*), Δ*speG*Δ*marA* and Δ*speG*Δ*marB* strains did not show any difference in growth compared to Δ*speG* strain under spermidine stress (*Figure 5E*). Note that, unlike Δ*speG* strain, the single mutants, viz. Δ*marA*, Δ*marB*, Δ*ygfZ*, Δ*iscU*, and Δ*soxS*, grow similarly to the WT strain in the presence or absence of spermidine (*Figure 5E* and *Figure 5—figure supplement 1*).

## Free spermidine interacts and oxidizes $Fe^{2+}$ ion to generate superoxide radicals in vitro

To probe whether spermidine directly interacts with iron, we performed isothermal titration calorimetry (ITC) using $Fe^{3+}$ (ferric citrate) and $Fe^{2+}$ (ferrous ammonium sulfate) ions. Titration of spermidine with $Fe^{3+}$ generated exothermic peaks indicating a standard binding reaction with a stoichiometry (N) of 0.711 (*Figure 6A*). On the other hand, titration of spermidine with $Fe^{2+}$ in two different isothermal conditions produced consistent and complex patterns (*Figure 6B and C*). To explain it, we divided the pattern into two halves. In the first half, $Fe^{2+}$ injections to spermidine generated alternate exothermic and endothermic peaks till the ratio of spermidine to $Fe^{2+}$ reaches about 1:1.3 (*Figure 6B and C*). In the second half of the profile, after the ratio of spermidine to $Fe^{2+}$ crosses 1:1.3, no endothermic peaks were observed, and a gradual shortening of exothermic peaks was generated, leading to saturation (*Figure 6B and C*). From the first half of pattern, we suspected $Fe^{2+}$ interaction with spermidine also involves some other reactions, such as oxidation of the $Fe^{2+}$ to generate $Fe^{3+}$ and $O_2^-$, $Fe^{3+}$ release, and subsequent $Fe^{3+}$ binding to spermidine.

To test whether $Fe^{2+}$ was oxidized in the presence of spermidine to liberate $Fe^{3+}$, we titrated spermidine by increasing amounts of $Fe^{2+}$ iron followed by assessing the level of $Fe^{2+}$ by using bipyridyl chelator. Chelation of $Fe^{2+}$ ions by bipyridyl generates pink color indicating $Fe^{2+}$ levels. No color formation was observed till the ratio of spermidine to $Fe^{2+}$ reaches 1:1.3 (*Figure 6D*), a number that

**Table 1.** The list of strains and plasmids used in this work.

| Strains and plasmids | Genotype/features | References |
|---|---|---|
| **Strains** | | |
| BW25113 | *Escherichia coli; rrnB3 ΔlacZ4787 hsdR514Δ(araBAD) 567 Δ(rhaBAD)568 rph-1* | *Baba et al., 2006* |
| Δ*speG* | BW25113, Δ*speG::kan^R* | *Baba et al., 2006* |
| Δ*sodA* | BW25113, Δ*sodA::kan^R* | *Baba et al., 2006* |
| Δ*sodB* | BW25113, Δ*sodB::kan^R* | *Baba et al., 2006* |
| Δ*zwf* | BW25113, Δ*sodC::kan^R* | *Baba et al., 2006* |
| Δ*fis* | BW25113, Δ*fis::kan^R* | *Baba et al., 2006* |
| Δ*ihfA* | BW25113, Δ*ihfA::kan^R* | *Baba et al., 2006* |
| Δ*iscU* | BW25113, Δ*iscU::kan^R* | *Baba et al., 2006* |
| Δ*ygfZ* | BW25113, Δ*ygfZ::kan^R* | *Baba et al., 2006* |
| Δ*soxS* | BW25113, Δ*soxS::kan^R* | *Baba et al., 2006* |
| Δ*marA* | BW25113, Δ*marA::kan^R* | *Baba et al., 2006* |
| Δ*marB* | BW25113, Δ*marB::kan^R* | *Baba et al., 2006* |
| Δ*ahpC* | BW25113, Δ*ahpC::kan^R* | *Baba et al., 2006* |
| Δ*katG* | BW25113, Δ*katG::kan^R* | *Baba et al., 2006* |
| Δ*speG*Δ*sodA* | BW25113, Δ*speG*, Δ*sodA::kan^R* | This study |
| Δ*speG*Δ*sodB* | BW25113, Δ*speG*, Δ*sodB::kan^R* | This study |
| Δ*speG*Δ*sodA*Δ*sodB* | BW25113, Δ*speG*, Δ*sodA*, Δ*sodB::kan^R* | This study |
| Δ*speG*Δ*zwf* | BW25113, Δ*speG*, Δ*zwf::kan^R* | This study |
| Δ*speG*Δ*soxS* | BW25113, Δ*speG*, Δ*soxS::kan^R* | This study |
| Δ*speG*Δ*fis* | BW25113, Δ*speG*, Δ*fis::kan^R* | This study |
| Δ*speG*Δ*ihfA* | BW25113, Δ*speG*, Δ*ihfA::kan^R* | This study |
| Δ*speG*Δ*iscU* | BW25113, Δ*speG*, Δ*iscU::kan^R* | This study |
| Δ*speG*Δ*ygfZ* | BW25113, Δ*speG*, Δ*ygfZ::kan^R* | This study |
| Δ*speG*Δ*marA* | BW25113, Δ*speG*, Δ*marA::kan^R* | This study |
| Δ*speG*Δ*marB* | BW25113, Δ*speG*, Δ*marB::kan^R* | This study |
| JRG3533 | MC4100 φ(*sodA-lacZ*)49, cm^R | *Tang et al., 2002* |
| RKM1 | BW25113, Δ*speG*, *sodA-lacZ*:cm^R | This study |
| **Plasmids** | | |
| pET28a (+) | *kan^R*; T7-promoter; IPTG inducible | Novagen |
| pBAD/Myc-His A | *amp^R*; pBAD-promoter; Ara inducible | ThermoFisher |
| pDAK1 | pBAD/*Myc*-His A; Two *Nde*I sites were mutated and *Nco*I site was replaced by *Nde*I | Lab resource |
| pZwf | *zwf* cloned in pDAK1 *Nde*I and *Hind*III sites | This study |
| pSodA | *sodA* cloned in pDAK1 vector | This study |
| pET-*sodA* | *sodA* cloned in pET28a (+) vector | This study |
| pET-*ahpC* | *ahpC* cloned in pET28a (+) vector | This study |
| pET-*katG* | *katG* cloned in pET28a (+) vector | This study |
| pSpeG | *speG* cloned in pDAK1 vector | This study |

*Table 1 continued on next page*

*Table 1 continued*

| Strains and plasmids | Genotype/features | References |
| --- | --- | --- |
| pUA66_soxS | $kan^R$; *soxS* promoter cloned upstream of *gfpmut2* reporter in pUA66 | *Zaslaver et al., 2006* |
| pUA66_ahpC | $kan^R$; *ahpC* promoter cloned upstream of *gfpmut2* reporter in pUA66 | *Zaslaver et al., 2006* |
| pUA66_katG | $kan^R$; *katG* promoter cloned upstream of *gfpmut2* reporter in pUA66 | *Zaslaver et al., 2006* |

Note: $kan^R$, kanamycin resistance; $amp^R$, ampicillin resistance, and $cm^R$, chloramphenicol resistance.

exactly matches with the ratio of spermidine to $Fe^{2+}$ in the first half of ITC experiments (*Figure 6B and C*). The color formation starts appearing when the ratio crosses 1:1.3 (*Figure 6D*), suggesting that 1 molecule of spermidine (or 10 molecules) exactly oxidizes 1.3 molecules (or 13 molecules) of $Fe^{2+}$. The colorimetric values overlap with the standard curve when reactions were under anoxic condition, indicating $Fe^{2+}$ was not oxidized (*Figure 6D*). We used nitro blue tetrazolium (NBT) dye to check whether the loss of one electron from $Fe^{2+}$ generates $O_2^-$ anion under spermidine stress. An increased NBT absorption at 575 nm till the ratio of spermidine to $Fe^{2+}$ reaches 1:1.3 confirms that 1 molecule (or 10 molecule) of spermidine interacts with 1.3 molecules (or 13 molecules) of $Fe^{2+}$ generating 1.3 molecules (or 13 molecules) $O_2^-$ anion radical (*Figure 6E*). From the stoichiometry of 0.711 (which is close to 0.5) (*Figure 6A*), we postulate that two spermidine and one $Fe^{3+}$ together could form a hexadentate coordination complex with an octahedral geometry (*Figure 6F*). It appears that when spermidine molecules engaged to form a hexadentate coordination complex with $Fe^{2+}$, the former helps oxidizing latter to form $Fe^{3+}$ in sufficient concentrations. $Fe^{3+}$ finally forms coordination complex with spermidine (*Figure 6F*). It may be noted that the binding of spermidine and $Fe^{3+}$ is entirely enthalpy-driven, as indicated by a large negative ΔH. The negative entropy (ΔS) value presumably results from the ordering of spermidine from an extended conformation to a compact and rigid one after metal chelation (*Figure 6A*).

The cellular spermidine barely exists as a 'free' species; rather, majority of them remain 'bound' with RNA, DNA, nucleotides, and phospholipids (*Igarashi and Kashiwagi, 2000*; *Miyamoto et al., 1993*; *Schuber, 1989*; *Tabor and Tabor, 1984*). It has been reported that these phosphate-containing biomolecules have the inherent property to inhibit iron oxidation blocking $O_2^-$ production (*LØVaas, 1996*; *Tadolini, 1988a*; *Tadolini, 1988b*). The bound spermidine further enhances the inhibitory effects of these biomolecules toward iron oxidation. Consistent with the report (*Tadolini, 1988b*), we noticed that 1 μg of RNA inhibited the oxidation of 200 μM $Fe^{2+}$. The presence of 10 μM spermidine further decreased iron oxidation (*Figure 6G*). However, increasing the concentrations of spermidine (50, 100, and 200 μM) accelerated iron oxidation gradually (*Figure 6G*). This data clearly indicates that cell maintains a level of cellular spermidine that may remain optimally bound with the biomolecules inhibiting $O_2^-$ generation. However, when homeostasis fails due to *speG* deletion, excess spermidine accumulates that can remain in a 'free' form inducing $O_2^-$ radical toxicity.

## Discussion

Our study presented in this paper answers why spermidine homeostasis is intriguingly fine-tuned in bacteria. We provide clear-cut evidence that excess spermidine, which remains as a free species (*Figure 6G*), stimulates the production of toxic levels of $O_2^-$ radicals in *E. coli* (*Figures 1 and 2*). $O_2^-$ anion thus generated affects cellular redox balance (*Figure 3*) and damages iron-sulfur clusters of the proteins (*Figures 4 and 5*). Since spermidine directly interacts with $Fe^{2+}$ (*Figure 6*), it may abstract iron from some of the iron-sulfur clusters, thereby inactivating some of the proteins. On the other hand, when spermidine level is at optimum, most of it remain as bound form with the biomolecules, thereby slows down iron oxidation and subsequent $O_2^-$ production (*LØVaas, 1996*; *Tadolini, 1988a*; *Tadolini, 1988b*; *Figure 6G*). Thus, spermidine deficiency would enhance the rate of iron oxidation (*Figure 6G*), leading to ROS production (*Figure 1A and B*). This is why spermidine is a double-edged sword where in excess, it provokes $O_2^-$ anion production, and in scarcity, it leads to higher ROS levels.

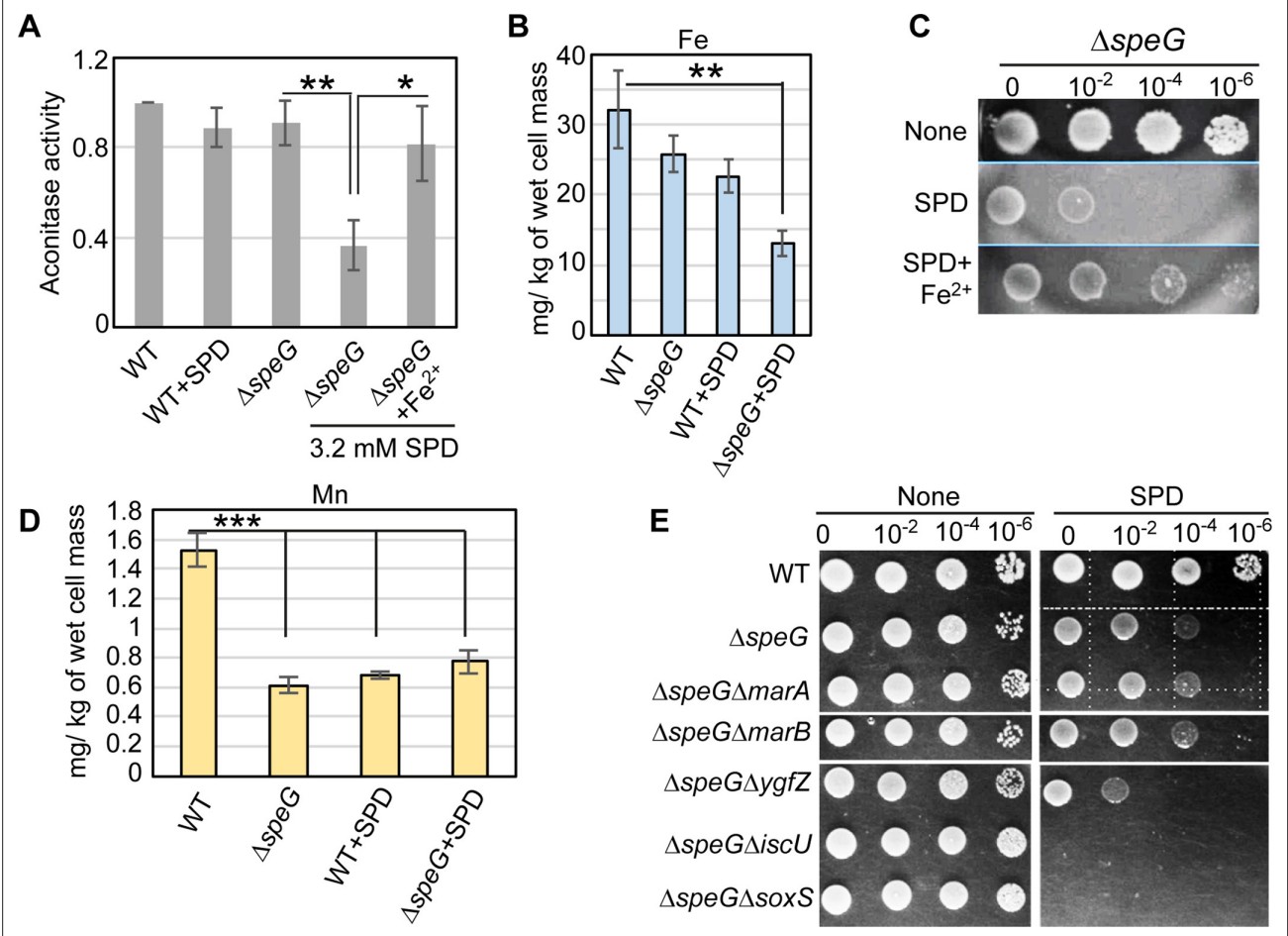

**Figure 5.** Spermidine-mediated $O_2^-$ radical production affects iron metabolism. (**A**) The bar diagram represents relative aconitase activity in the *Escherichia coli* WT and Δ*speG* strains in the presence and absence of spermidine (SPD). (**B**) Intracellular levels of Fe in the *E. coli* strains determined in the presence or absence of SPD stress were plotted. (**C**) Spot assay using serially diluted Δ*speG* cells demonstrated that $Fe^{2+}$ can rescue SPD stress. (**D**) Intracellular levels of Mn levels in the *E. coli* strains determined in the presence or absence of SPD stress were plotted. (**E**) Spot assay shows the relative sensitivity of various double mutants, Δ*speG*Δ*ygfZ*, Δ*speG*Δ*iscU*, and Δ*speG*Δ*soxS* strains to SPD. Error bars in the panels are mean ± SD from the three independent experiments. Whenever mentioned, the ***, **, and * denote p-values < 0.001, < 0.01, and < 0.1 respectively; unpaired t test. See also *Figure 5—figure supplement 1*, and *Figure 5—source data 1*, *Figure 5—source data 2*, *Figure 5—source data 3*.

The online version of this article includes the following source data and figure supplement(s) for figure 5:

**Source data 1.** *Figure 5A* Raw data.

**Source data 2.** *Figure 5B* Raw data.

**Source data 3.** *Figure 5D* Raw data.

**Figure supplement 1.** Spermidine (SPD) sensitivity of the single mutants.

Polyamines remain protonated at physiological pH, yet they are able to coordinate several positively charged metal ions, such as $Ni^{2+}$, $Co^{2+}$, $Cu^{2+}$, and $Zn^{2+}$, possibly via charge neutralization by counterions that reduces the Coulombic repulsion between spermidine and the metals (*LØVaas, 1996*). Similar charge neutralization of the nitrogen atoms of spermidine likely allows coordinate covalent bonds with $Fe^{3+}$ (*Figure 6F*). About 10 spermidine molecules oxidize $Fe^{2+}$ to generate 13 $Fe^{3+}$ cations and equivalent numbers of $O_2^-$ radicals (*Figure 6B and C*). When sufficient concentration of $Fe^{3+}$ is generated, two spermidine molecules coordinate one $Fe^{3+}$ to form a hexadentate complex with an octahedral geometry (*Figure 6F*). We substantiated this in vitro spermidine-mediated iron oxidation and subsequent $O_2^-$ radical production phenomena (*Figure 6*), showing that cells are highly toxic to the spermidine under aerobic condition but not under anaerobic condition (*Figure 1D*).

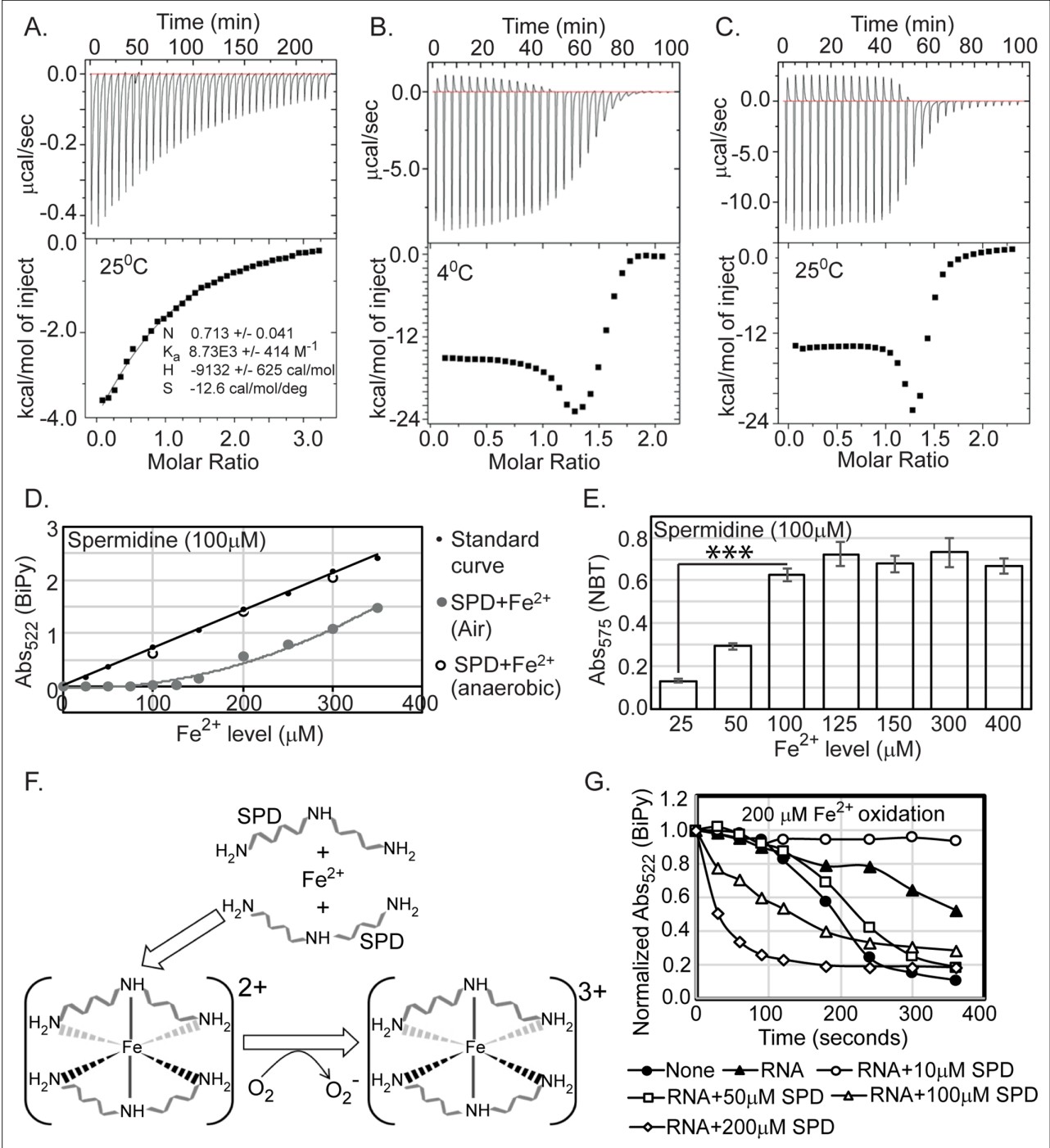

**Figure 6.** Spermidine oxidizes $Fe^{2+}$ generating $O_2^-$ radical in aerobic condition. (**A**) Isothermal titration calorimetry (ITC) data demonstrates the interaction of spermidine with $Fe^{3+}$. (**B**) and (**C**) ITC data shows the interaction of spermidine with $Fe^{2+}$ ion at 4°C and 25°C, respectively. (**D**) 100 μM spermidine was incubated with different concentrations of $Fe^{2+}$ followed by estimation of $Fe^{2+}$ levels by bipyridyl chelator. The color formation was recorded at 522 nm and plotted them along with standard curve. The panel depicts that the incubations of 100 μM spermidine with 100, 200, and 300 μM of $Fe^{2+}$ in the anaerobic condition do not lead to the loss of $Fe^{2+}$ ions detected by bipyridyl chelator. However, when 100 μM spermidine was incubated with the different concentrations of $Fe^{2+}$ (25–350 μM) in the aerobic condition, the bipyridyl-mediated color formation was observed when $Fe^{2+}$ level was between above 125 μM and 150 μM (i.e., till spermidine to $Fe^{2+}$ ratio reaches approximately 1.3). The mean values from the three independent experiments were plotted. SD is negligible and is not shown for clarity. (**E**) Nitro blue tetrazolium (NBT) assay was performed to determine that spermidine and $Fe^{2+}$ interaction yields $O_2^-$ radical. The colorimetry at 575 nm suggests that 100 μM of spermidine interacts with approximately 125 μM of $Fe^{2+}$ (ratio 1:1.3) to generate saturated color. Error bars in the panel are mean ± SD from the three independent experiments. *** denotes p-value < 0.001; unpaired t test. (**F**) Model to show final coordination complex formation. An $Fe^{2+}$ interacts with two spermidine molecules

*Figure 6 continued on next page*

*Figure 6 continued*

forming hexadentate coordination complex. This interaction oxidizes $Fe^{2+}$ liberating one electron to reduce oxygen molecule. Finally, two spermidine coordinates one $Fe^{3+}$ with an octahedral geometry. (**G**) The curves represent the *Escherichia coli* total RNA inhibits iron oxidation. Spermidine further reduces the RNA-mediated iron oxidation at concentration 10 µM but higher concentrations of spermidine increase the iron oxidation despite the presence of RNA. The mean values are derived from the three independent experiments and plotted. SD is negligible and is not shown for clarity. See also *Figure 6—source data 1*, *Figure 6—source data 2*, *Figure 6—source data 3*.

The online version of this article includes the following source data for figure 6:

**Source data 1.** *Figure 6D* Raw data.

**Source data 2.** *Figure 6E* Raw data.

**Source data 3.** *Figure 6G* Raw data.

Usually, abundant $O_2^-$ level leads to general ROS including $H_2O_2$ production. However, despite elevated $O_2^-$ production, spermidine lowers overall ROS levels in Δ*speG* strain (*Figures 1C, A and 2*). The declined $H_2O_2$ level could be attributed to the slower rate of $O_2^-$ anion dismutation due to the failure of *sodA* activation (*Figure 4—figure supplement 2*) and the activation of alkyl hydroperoxidase (AhpCF) that neutralizes $H_2O_2$, represented by AhpC overexpression (*Figure 4E*). A low level of cellular manganese (*Figure 5D*) could also limit SodA activity. Besides, the activation of IscR regulon (*Figure 4A*), the low cellular iron content (*Figure 5B*), and the rejuvenation of cell growth by $Fe^{2+}$ supplementation (*Figure 5C*) indicate that the spermidine presumably lowers the $Fe^{2+}/Fe^{3+}$ ratio in Δ*speG* strain. Thus, the decreased level of $Fe^{2+}$ and $H_2O_2$ (*Figures 5B and 1C*) could potentially diminish cellular •OH radical production in the spermidine-fed cells. We have summarized all these observations and hypotheses in the schematic *Figure 7*.

Interestingly, spermidine stimulates $O_2^-$ production but SoxR function remained indifferent in the Δ*speG* cells (*Figure 4B*). This observation is consistent with the previous finding that redox cycling drugs, but not $O_2^-$, are the efficient activators of SoxR function (*Gu and Imlay, 2011*). Even spermidine blocked SoxR expression by menadione, a redox cycling drug (*Figure 4B*). These two observations implicate that free spermidine being an iron chelator (*Figure 6A, B and C*) might affect SoxR maturation by interfering its iron-sulfur cluster formation. As a result, apo-SoxR remained unreactive to the superoxide or redox cycling drugs, and thereby failed to activate SoxR regulon genes. Since spermidine ubiquitously interacts with DNA and modulates gene expression in many ways (*Igarashi and Kashiwagi, 2000*; *Jung and Kim, 2003*; *Miyamoto et al., 1993*), another possibility could be that excess of it might occlude SoxR-binding to the *soxS* and *sodA* promoter regions to activate them. Alternatively, blockage of SoxR activation could result from spermidine-mediated activation of *rsxA* and *rsxB* (*Figure 4A*), which encode the critical components of the iron-sulfur cluster reducing system of SoxR (*Koo et al., 2003*), to keep SoxR inactive. Nevertheless, a detailed biochemical study on this aspect is needed to understand the mechanism.

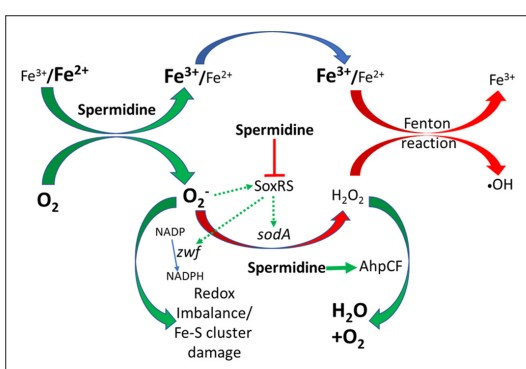

**Figure 7.** Flowchart explaining the reactive oxygen species (ROS) generation under spermidine stress. The model describes that the spermidine administration in the cell interacts with free iron and oxygen to generate $O_2^-$ radical, increasing $Fe^{3+}/Fe^{2+}$ ratio. Spermidine also blocks $O_2^-$ radical-mediated activation of SoxRS that upregulates *zwf* and *sodA*. Consequently, reduced nicotinamide adenine dinucleotide phosphate (NADPH) production and dismutation of $O_2^-$ radical to $H_2O_2$ were not accelerated, leading to redox imbalance and $O_2^-$-mediated damage to the iron-sulfur clusters, respectively. Additionally, spermidine translationally upregulated alkyl hydroperoxidase (AhpCF) that lowers the level of $H_2O_2$. Declined cellular $Fe^{2+}$ and $H_2O_2$ levels weaken Fenton reaction to produce •OH radical.

Our study in *E. coli* observed quite a few biochemical aspects which might explain how the horizontal acquisition of *speG* gene could confer a pathogenic advantage to the *Staphylococcus aureus* USA 300 strain (*Eisenberg et al., 2009*). *S. aureus*, a Gram-positive commensal living on human skin, often causes severe disease upon access to deeper tissues. Since most of the iron in mammals exists intracellularly, the extracellular pathogen, *S. aureus* faces hardship and competes

with the host for the available iron (*Hammer and Skaar, 2011*). As spermidine declines cellular iron content and interferes with iron metabolism (*Figure 4*), it is thus possible that *S. aureus* does not synthesize spermidine (*Joshi, 2012*). Furthermore, the acquisition of *speG* gene by S. *aureus* USA300 (*Joshi, 2012*) could allow it to inactivate host-originated spermidine/spermine, thereby to maintain cellular iron content. Corroborating to our findings, a recent observation has pointed out that spermine stress upregulates iron homeostasis genes, indicating that spermine toxicity has a specific connection with iron depletion in the *speG*-negative *S. aureus* strain, Mu50 (MRSA) (*Yao and Lu, 2014*). Besides, spermine-mediated iron depletion may be responsible for the synergistic effect of spermine with the antibiotics against *S. aureus* (*Kwon and Lu, 2007*). Nevertheless, a thorough in vivo host-pathogen interaction study may unravel a specific link between spermine/spermidine and iron depletion in *S. aureus*.

## Materials and methods

### Bacterial strains, plasmids, proteins, and chemicals

Bacterial strains and plasmids used in this study are listed in *Table 1*. BW25113 strain of *E. coli* was used as WT in this study. Oligonucleotides were purchased from IDT. Bacterial broths and agar media were purchased from BD Difco. The knockout strains of *E. coli* were procured from the KEIO library (*Baba et al., 2006*), verified by PCR, freshly transduced into the WT background by P1 phage, and sequenced to confirm the deletion. The double and triple knockout mutants were generated following the standard procedure described by *Datsenko and Wanner, 2000*. *E. coli* strain JRG3533 was a generous gift from Dr Rachna Chaba, IISER Mohali, India. RKM1 strain was constructed by P1 transduction of *sodA-lacZ*:Cm^R genotype of JRG3533 to BW25113Δ*soxS* strain.

The plasmids, pUA66_*soxS*, pUA66_*ahpC*, pUA66_*katG,* were the gifts from Dr Csaba Pal, Biological Research Centre of the Hungarian Academy of Sciences (*Zaslaver et al., 2006*). pBAD-*zwf* was a generous gift from Dr CC. Vasquez, Universidad de Santiago de Chile (*Sandoval et al., 2011*). *sodA*, *katG*, *ahpC*, and *speG* genes were PCR-amplified by DG12-DG13, RM7-RM8, DG9-DG10, and RK3-RK4 primer pairs (*Supplementary file 3*), respectively. The PCR products were double-digested at the primer-specific unique restriction sites and inserted into identically digested pET28a (+) plasmid vector so that the 6X His-tagged SodA, KatG, and AhpC proteins are being produced (*Table 1*). The protein expression vectors, pET-*sodA*, pET-*ahpC*, pET-*katG,* were transformed to BL21 (DE3) cells, and expressions were induced by 0.4 mM IPTG. The overexpressed proteins were purified using Ni-NTA beads. The purified proteins were used to raise rabbit polyclonal antibodies following the standard procedure. *sodA* and *speG* were additionally subcloned in pDAK1, a derivative of pBAD/Myc-His A vector to get pSodA, and pSpeG multicopy expressions for complementation assays (*Table 1*). We also PCR-amplified *zwf* using RK55-RK56 primer pairs and cloned in the pDAK1 vector to get pZwf vector for complementation assays.

### Growth, viability, spermidine sensitivity, and complementation assays

An automated BioscreenC growth analyzer (Oy growth curves Ab Ltd) was used to generate growth curves mentioned in the Results. For this purpose, overnight cultures of different strains were diluted in fresh LB medium and grown in the presence and absence of 3.2–6.4 mM of spermidine. Ten mM of each of the ROS quenchers (TU, Tr, SP, ascorbate, and NAC) were used wherever mentioned. For growth assay of *E. coli* strains on the LB-agar supplemented with or without spermidine were performed by spotting serially diluted overnight cultures and growing them at 37°C. For viability assays, serially diluted *E. coli* strains were spread on LB-agar surface supplemented with 6.4 mM spermidine. We determined the viability under spermidine stress from the number of colonies grown. ZOIs, which appeared following overnight growth of the strains in the presence of 6.4 mM spermidine in the wells on agar plates, were determined both in aerobic and in anaerobic conditions. The anaerobic condition was created in an anaerobic Petri dish jar using AnaeroGas Pack 3.5 l pouches. For complementation experiments, the pSodA, pSpeG, and pZwf plasmids were transformed into Δ*speG*Δ*zwf* and Δ*speG*Δ*sodA* strains, respectively, and growth assays were performed in the presence of spermidine. Since the leaky expressions were sufficient to rescue growth defects, induction with arabinose was avoided for this purpose.

The reporter plasmids, pUA66_soxS, pUA66_ahpC, pUA66_katG, were transformed into ΔspeG strain. The transformed cells were grown in the presence or absence of 3.2 mM spermidine. Wherever mentioned, 25 µM menadione was used as a positive control for $O_2^-$ generation. The cell pellets were washed twice with PBS and dissolved in 500 µl phosphate buffer saline (PBS). Flow cytometry was done using the Fl1 laser for 0.05 million cells using FACSVerse (BD Biosciences). The MFI values from three biological replicates have been calculated.

## Determining relative ROS levels in the cells

H2DCFDA (10 µM) and DHE (2.5 µM) were used to measure cellular •OH and $O_2^-$ anion, respectively. The cells were grown in the presence or absence of 3.2 mM spermidine. Cells were harvested, washed with PBS, and an equal mass of cell pellets was incubated with DHE or H2DCFDA probes for an hour. The data were acquired using BD accuri Fl3 laser (for DHE) and Fl1 laser (for H2DCFDA) for 0.05 million cells. The MFI values of triplicate experiments were calculated. For $H_2O_2$ detection, the *E. coli* cells were grown in the presence or absence of 3.2 mM of spermidine for 4 hr. Cells were harvested and washed with 1× M9 minimal media. The equal mass of cells (2.5 mg each) suspended in 6 ml M9 minimal media were incubated for different time points to allow $H_2O_2$ liberation. The relative $H_2O_2$ liberation was measured by a Fluorimetric Hydrogen Peroxide Assay kit (Sigma Aldrich).

## EPR spectroscopy

The protocol was adopted from *Thomas et al., 2015*, with some modifications. The ΔspeG strain harboring pDAK1 empty vector or pSodA was grown in the presence or absence of 3.2 mM spermidine for 2 hr and then 0.001% arabinose was added and further grown for 2 more hours; 100 mg cell pellets were quickly resuspended in 700 µl of KDD buffer, pH 7.4 (99 mM NaCl, 4.69 mM KCl, 2.5 mM $CaCl_2$, 1.2 mM $MgSO_4$, 25 mM $NaHCO_3$, 1.03 mM $KH_2PO_4$, 5.6 mM D-glucose, 20 mM HEPES, 5 µM DETC, and 25 µM deferoxamine); 100 µl cell suspensions were preincubated with or without 20 mM DMTU and 200 µM UA for 5 min; and 500 µM of CMH spin probe (Enzo Life Sciences) were added and incubated for 30 min at 37°C. EPR spectra were acquired using a Bruker EMX MicroX EPR spectrometer with the following settings: center field, 3438 G, sweep width, 500 G; microwave frequency, 9.45 GHz; microwave power, 8.04 mW; modulation frequency 100 kHz; modulation amplitude, 5.64 G; conversion time, 40 ms; time constant, 40.96 ms; receiver gain, 1120; data points 1024; number of X-Scans, 5.

## β-Galactosidase and GFP reporter assays

For the β-galactosidase assay, the RKM1 strain was grown in the presence or absence of 3.2 mM of spermidine. The cell pellets were washed twice with Z-buffer (60 mM $Na_2HPO_4$, 40 mM $NaH_2PO_4$, 10 mM KCl, and 1 mM $MgSO_4$) and diluted to $OD_{600}$ ~0.5. Promoter activity was measured by monitoring β-galactosidase expression from single-copy *sodA-lacZ* transcriptional fusion; 100 µl of 4 mg/ml ONPG was used as a substrate, which was cleaved by β-galactosidase to produce yellow-colored *O*-nitrophenol. Colorimetric detection of this compound was done at 420 nm.

The reporter plasmids, pUA66_soxS, pUA66_ahpC, pUA66_katG, containing GFP-mut2 reporters, were used to determine the promoter activities of *soxS*, *ahpC*, and *katG* genes in the presence or absence of 3.2 mM spermidine. Flow cytometry was done using the FL1 laser for 0.05 million cells using FACSVerse (BD Biosciences) or BD Accuri C6 Plus Flow Cytometer (BD Biosciences) machine.

## Western blotting experiments

Overnight culture of *E. coli* strains was inoculated in fresh LB medium in 1:100 dilution and grown for 1.5 hr at 37°C. Next, 3.2 mM of spermidine were added, wherever required and allowed to grow again at 37°C for 2.5 hr. Cells were harvested and lysed with B-PER bacterial protein extraction reagent (Thermo Scientific). The total protein level was checked by the Bradford assay kit (Bio-Rad); 40 µg of total cellular proteins from the individual samples were subjected to SDS-PAGE. The proteins were transferred to a nitrocellulose membrane and stained with Ponceau S to visualize protein resolution and equal loading in the PAGE. Western blotting was performed using polyclonal rabbit primary antibodies and HRP-conjugated secondary antibodies. The blots were developed by Immobilon Forte Western HRP substrate (Millipore).

## Estimating cellular spermidine levels

Cells were grown in presence or absence of spermidine for 4 hr. The cells were washed with 1 M NaCl at 37°C for 10 min; 500 nmol of hexane-diamine (internal standard) was added and the pellets were resuspended in 750 µl of 10% perchloric acid. The cells were lysed by freeze-thawing using liquid nitrogen, and 800 µl of saturated sodium carbonate and 800 µl of 10 mg/ml of dansyl chloride were added to the supernatants. The dansylation was carried out at 60°C for 3 hr in dark. The reaction was stopped using 400 µl of 100 mg/ml proline and kept at 60°C for 30 min; 400 µl toluene was added to each sample and mixed thoroughly. The organic layer was collected and dried using a speed vac; 2 ml 80% acetonitrile was added and sonicated to dissolve the dry samples. The samples were then passed through 0.22 µm filter and injected to HPLC system (Agilent 1260 Infinity II) attached with a reversed-phase C-18 column (Agilent ZORBAX Eclipse Plus C18 of dimension 4.6 × 100 mm, 3.5 µm). Acetonitrile gradient (0–100%) with 0.8 ml/min flow rate was used for all samples. A PDA detector was used to monitor the elution peaks. The corresponding mass of individual peaks were detected using either a single quadrupole Agilent MSD using the ESI source or a separate Agilent LC-MS/MS equipment. Pure spermidine and hexane-diamine were also dansylated and determined their 100% tri- or di-dansylation. The dansylated spermidine was also used to generate a standard curve. The peak areas of spermidine (mAu*s) were normalized with the average peak area of internal standards. The absolute amounts of spermidine were calculated from the standard curve.

## Isothermal titration calorimetry

A MicroCal VP-ITC calorimeter, MicroCal Inc, was used for calorimetric measurements to probe the interaction of spermidine with $Fe^{2+}$ and $Fe^{3+}$ species. In order to achieve this, 100 µM of spermidine solution was prepared in 20 mM sodium acetate buffer (pH 5.5) and put into the sample cell. The ligands, 2.1 mM of $FeCl_3$ or ferrous ammonium sulfate, were also dissolved in the identical sodium acetate buffer. The titrations involved 30 injections of individual ligands (5 µl per shot) at 300 s intervals into the sample cell containing 1.8 ml of 100 µM spermidine. The titration cell was kept at some specific temperature and stirred continuously at 286 rpm. The heat of dilution of ligand in the buffer alone was subtracted from the titration data. The data were analyzed using Origin 5.0 software.

## 2,2′-Bipyridyl and NBT assays

2,2′-Bipyridyl chelates $Fe^{2+}$ producing color that absorbs at 522 nm ($A_{522}$). The standard curve for 0–350 µM of $Fe^{2+}$ ion was generated simply by recording $A_{522}$ in the presence of 2,2′-bipyridyl. Dissolved oxygen of medium and headspace oxygen was replaced by flushing $N_2$ gas in the medium for 5 min to create an anoxic condition as described (*Stieglmeier et al., 2009*). To check whether spermidine acts as a catalyst for $Fe^{2+}$ to $Fe^{3+}$ oxidation, we performed 2,2′-bipyridyl assay probing leftover $Fe^{2+}$ after the reaction. For this assay, 100 µM of spermidine was incubated with increasing concentrations (25–350 µM) of ferrous ammonium sulfate for 10 min at room temperature (RT); 900 µl of the reaction products were mixed with 90 µl 4 M sodium acetate buffer (pH 4.75) and 90 µl bipyridyl (0.5% in 0.1 N HCl). The color formation was recorded at 522 nm ($A_{522}$) using UV-1800 Shimandzu UV-spectrophotometer. In another experiment, the assay was performed in anoxic condition using rubber-capped sealed glass vials containing anoxic reactants and needle-syringe-mediated mixing of the reagents. Here, three different concentrations (100, 200, and 300 µM) of ferrous ammonium sulfate were reacted with 100 µM of spermidine for 10 min followed by spectrophotometry at $A_{522}$. The standard curve for 0–350 µM of $Fe^{2+}$ ion was generated simply by recording $A_{522}$ of the mixture of 900 µl ferrous ammonium sulfate, 90 µl sodium acetate buffer, and 90 µl bipyridyl solutions.

Iron oxidation in the presence of RNA and spermidine was performed as described (*Tadolini, 1988b*). One µg RNA and increasing concentrations of spermidine (10–200 µM) were used in 5 mM MOPS buffer, pH 7.4. The oxidation was started adding 200 µM $FeCl_2$. The reactions were stopped at desired time point by adding a stop solution (1:1 4 M sodium acetate:4 M glacial acetic acid) followed by 2,2′-bipyridyl to detect $Fe^{2+}$ levels.

We used NBT dye to probe whether spermidine-stimulated $Fe^{2+}$ to $Fe^{3+}$ oxidation liberates $O_2^-$ anion in vitro. For this assay, different concentrations of $Fe^{2+}$ were incubated with 100 µM of spermidine for 2 min; 100 µl of NBT (5 mg/ml) was added to the mixture and incubated at RT for another 5 min. The absorbance was recorded at 575 nm using UV-1800 Shimandzu UV-spectrophotometer.

## RT-qPCR

Bacterial mRNAs were isolated by TRIzol reagent and the Qiagen bacterial RNA isolation Kit. DNase I treatment was done to remove residual DNA contaminant, and the integrity of the mRNA was checked on a 1% agarose gel. The RNA concentration was determined by a Nano-drop spectrophotometer (Thermo Scientific) and by a UV-1800 Shimandzu UV-spectrophotometer; 200 ng of RNA samples, primer pairs (*Supplementary file 3*), and GoTaq 1-Step RT-qPCR System (Promega) were used for RT-qPCR. Reaction mixture without template were included as negative controls. At least three independent experiments were conducted for the determination of cycle threshold ($C_T$) values. Fold expression change between spermidine-fed and unfed samples was calculated by the $\Delta\Delta C_T$ method. The values were normalized to the level of *betB* mRNA that was expressed constitutively as observed in the microarray.

## Other biochemical assays

The relative levels of cellular NAD$^+$/NADH and NADP/NADPH were measured using MAK037 and MAK038 kits (Sigma), respectively. ATP Bioluminescence assay Kit CLS II (Roche) were used to determine cellular ATP levels. The glutathione assay was performed, as described (*Rahman et al., 2006*). Cells were grown in the presence or absence of 3.2 mM spermidine for 4 hr. The PBS-washed cell pellets were kept on the ice.

For NAD$^+$/NADH and NADP/NADPH assays, 30 mg of cell pellets were dissolved in 400 µl of extraction buffer supplemented with 50 µg/ml of lysozyme and sonicated. The supernatants were collected and passed through 10 kDa spin columns; 10 µl of 0.1 N HCl or 0.1 N NaOH were added slowly for NAD$^+$ or NADH levels, respectively. On the other hand, 10 µl of 0.1 N NaOH or 0.1 N HCl were added slowly for NADP or NADPH levels, respectively. The samples were incubated at 60°C for 50 min; 50 µl of samples were mixed with the kit-specific 98 µl cycling buffer, and 2 µl cycling enzyme mix, and incubated at RT for 1 hr. Then, 10 µl of NADH or NADPH developer substrates were added in dark. A$_{450}$ were recorded and the colorimetric values were directly used to calculate the relative levels of the individual species.

For ATP estimation, 30 mg cell pellets were resuspended in 100 mM Tris-HCl (pH 7.75), 4 mM EDTA, and then incubated in boiling water for 2 min. The supernatants were collected and kept on ice; 50 µl of the supernatants and 50 µl of luciferase reagent were mixed taken in 96-well, flat-bottom black microwell plate. Luminescence was measured using BIOTEK plate reader and the values were directly used to represent relative ATP levels.

For GS$_t$ assay, 20 mg *E. coli* cell pellets were resuspended in 5% sulfosalicylic acid and boiled at 95°C for 5 min; 100 µl supernatant was mixed with 700 µl KPE buffer, 0.6 mM DTNB, and 0.3 units of glutathione reductase; 0.2 mM of β-NADPH was added finally. To estimate oxidized form of glutathione (GSSG) only, the cell extracts were pretreated with 10 mM 2-vinylpyridine for 1 hr so that GSH were cross-linked with it. The excess 2-vinylpyridine was neutralized with tri-ethanolamine. The reactions were carried out for 30 min and A$_{412}$ was recorded and the values were directly used to calculate the relative levels of each species. Aconitase assay was performed as per the protocol described (*Gardner and Fridovich, 1992*). Metal contents were determined by ICP-MS analyses at Punjab Biotechnology Incubator, Mohali, India. The metal concentration in the cell was determined as parts per billion (mg/kg) of *E. coli* cell pellets.

## Microarray experiments and interpretation

The saturated overnight culture of Δ*speG* strain was inoculated in the fresh LB medium and grown for 1.5 hr. After that 3.8 mM spermidine was added to one of the flasks, and the cultures were grown further for 2.5 hr. The cell pellets were harvested and washed with PBS, and dissolved in RLT buffer. The microarray was done from Genotypic Technology, Bangalore. The microarray had three probes for each gene on average.

## RNA extraction and RNA quality control for microarray

*E. coli* cell pellet was resuspended in 300 µl of 5 mg/ml lysozyme and incubated at RT for 30 min. Isolation of RNA from *E. coli* was carried out using Qiagen RNeasy mini kit (Cat # 74106) as per manufacturer's guidelines. A separate DNase treatment of the isolated total RNA was performed. The purity of the RNA was assessed using the Nanodrop Spectrophotometer (Thermo Scientific; ND-1000), and

the integrity of the RNA was analyzed on the Bioanalyzer (Agilent 2100). We considered RNA to be of good quality based on the 260/280 values (Nanodrop), rRNA 28S/18S ratios, and RNA integrity number (RIN) (Bioanalyzer).

## Microarray labeling

The sample labeling was performed using Quick-Amp Labeling Kit, One Color (Agilent Technologies, Part Number: 5190-0442); 500 ng of each sample were denatured along with WT primer with a T7 polymerase promoter. The cDNA master mix was added to the denatured RNA sample and incubated at 40°C for 2 hr for double-stranded cDNA synthesis. Synthesized double-stranded cDNA was used as a template for cRNA generation. cRNA was generated by in vitro transcription, and the cyanine-3-CTP (Cy3-CTP) dye incorporated during this step and incubated at 40°C for 2.30 hr. The Cy3-CTP labeled cRNA sample was purified using the Qiagen RNeasy column (Qiagen, Cat # 74106). The concentration of cRNA and dye incorporation was determined using Nanodrop-1000.

## Microarray hybridization and scanning

About 4 µg of labeled Cy-3-CTP cRNA was fragmented at 60°C for 30 min, and the reaction was stopped by adding 2× GE HI-RPM hybridization buffer (Agilent Technologies, In situ Hybridization kit, Part Number: 5190-0404). The hybridization was carried out in Agilent's Surehyb Chambers at 65°C for 16 hr. The hybridized slides were washed using Gene Expression Wash Buffer 1 (Agilent Technologies, Part Number: 5188-5325) and Gene Expression Wash Buffer 2 (Agilent Technologies, Part Number: 5188-5326) and were scanned using Agilent Scanner (Agilent Technologies, Part Number: G2600D). Data extraction from the images was done using Feature Extraction Software Version 11.5.1.1 of Agilent.

## Microarray data analysis

Microarray data analysis was undertaken by in-house coded R Script (https://cran.r-project.org/). Processing of raw data into expression profiles was achieved by utilizing the packages limma and affy. Probe intensities were converted into expression measures by standard procedures. Briefly, the design-sets depicting the 'control/test' arrays were carefully generated by reading the raw data from MA image files. Background correction was done by the method 'normexp'. This data was quantile normalized (between arrays depending on the design set), and within-array replicates were averaged. Processed data were categorized into major functional categories and tabulated. The detailed microarray array discussed in this manuscript have been deposited in GEO with accession number GSE154618.

## Acknowledgements

The authors are grateful to Dr Debashish Adhikari, Division of Chemical Sciences, IISER Mohali, for their critical inputs on the plausible binding mechanism of spermidine and iron; to Prof. Kaushik Ghosh and Dr JS Meena, IIT Roorkee, India, and Mr LM Jha, IISER Bhopal, India, for their sincere support in EPR analyses. The work has been funded by CSIR IMTECH, India to DD. VK was an ICMR fellow, RKM and PD is a UGC fellow, DG is a CSIR fellow, AK is a DST-Inspire fellow, and AP is a DBT fellow.

## Additional information

### Funding

| Funder | Grant reference number | Author |
| --- | --- | --- |
| Council of Scientific and Industrial Research (CSIR), India | MLP-042 | Dipak Dutta |

The funders had no role in study design, data collection and interpretation, or the decision to submit the work for publication.

## Author contributions
Vineet Kumar, Investigation, Methodology, Validation, Writing – original draft; Rajesh Kumar Mishra, Formal analysis, Investigation, Methodology, Validation; Debarghya Ghose, Arunima Kalita, Investigation, Methodology, Validation; Pulkit Dhiman, Anand Prakash, Nirja Thakur, Gopa Mitra, Investigation, Methodology; Vinod D Chaudhari, Investigation, Methodology, Supervision; Amit Arora, Formal analysis, Investigation, Methodology; Dipak Dutta, Conceptualization, Funding acquisition, Investigation, Methodology, Project administration, Supervision, Validation, Writing – review and editing

## Author ORCIDs
Amit Arora ⓘ http://orcid.org/0000-0002-3503-4695
Dipak Dutta ⓘ http://orcid.org/0000-0002-0458-4109

## Ethics
Polyclonal antibodies were raised using NZW rabbits in an in-house animal facility. This animal handling was approved by the Institutional Animal Ethics Committee (IAEC) and performed according to the National regulatory guidelines issued by Committee for the Purpose of Supervision of Experiments on Animals (CPSEA), Govt. of India.

## Decision letter and Author response
Decision letter https://doi.org/10.7554/eLife.77704.sa1
Author response https://doi.org/10.7554/eLife.77704.sa2

## Additional files

### Supplementary files
• Supplementary file 1. Table listing microarray data representing upregulated (green) and downregulated (red) genes.
• Supplementary file 2. Table listing the Fis- and IHF-regulated genes that were upregulated (green) and downregulated (red).
• Supplementary file 3. Table listing the oligonucleotide primers used in this study.
• Transparent reporting form

### Data availability
Microarray data is available in the GEO server. GEO accession Number GSE154618 has been provided in the material and method section. Source files for the following Figures were provided as a zip folder: Figure 1A, 1B, 1C, 1F Figure 2 Figure 3A, 3B, 3C, 3D, 3E, 3F, 3G Figure 4B (ii), 4C, 4D, 4E Figure 5A, 5B, 5D Figure 6D, 6E, 6G Figure 1-figure supplement 1C.

The following dataset was generated:

| Author(s) | Year | Dataset title | Dataset URL | Database and Identifier |
|---|---|---|---|---|
| Dutta D, Kumar V, Mishra R, Arora A | 2021 | The global transcriptomic profile in the spermidine-stressed *E. coli* | https://www.ncbi.nlm.nih.gov/geo/query/acc.cgi?acc=GSE154618 | NCBI Gene Expression Omnibus, GSE154618 |

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
