## [Editor Report]

The authors argue that a polyamine, spermidine, causes the production of reactive oxygen species (ROS) in *Escherichia coli* by oxidizing Fe2+, but spermidine can also be protective against ROS at lower concentrations when bound to other cellular molecules such as RNA. Thus, spermidine has both protective and antagonistic effects on ROS stress, depending on the cellular concentration.

---

## [Decision Letter]

**Decision letter after peer review:**

[Editors’ note: the authors submitted for reconsideration following the decision after peer review. What follows is the decision letter after the first round of review.]

Thank you for submitting the paper "Free spermidine evokes superoxide radicals that manifest toxicity" for consideration by *eLife*. Your article has been reviewed by 3 peer reviewers, including Joseph T Wade as the Reviewing Editor and Reviewer #1, and the evaluation has been overseen by a Senior Editor.

Comments to the Authors:

We are sorry to say that, after consultation with the reviewers, we have decided to reject the paper.

The reviewers raised four major concerns. First, the probes used to determine the type of ROS stress are not specific enough to confidently draw the conclusion that spermidine causes O2- production, as opposed to another ROS. Hence, we would want to see a more direct test of the specific ROS produced. Electron paramagnetic resonance would be a suitable approach for this experiment. Second, several of the experiments use deletion mutant strains but lack a control where the deletion has been complemented, together with a corresponding empty-vector control. Third, the reviewers felt that the *S. aureus* experiments are peripheral to the main theme of the paper, and should be removed. Fourth, the reviewers were concerned that intracellular spermidine levels for different strains/growth conditions were inferred but not directly measured, particularly for the experiments in Figure 1 with speE and speG mutants and with spermidine supplementation. The use of LB as a growth medium is a concern in this regard, since LB contains spermidine. We would want to see either measurement of intracellular spermidine levels for wild-type and mutant strains grown under the conditions used in the corresponding experiments, or references to prior work where spermidine levels have been measured under the same growth conditions for the same strains. We believe these concerns are addressable, but would require a considerable amount of work, which would take more than two months. Hence, we are rejecting the paper, but we would consider a revised version if you are able to address the four major concerns.

*Reviewer #1:*

The work presented in this study provides important insight into the role of spermidine in bacterial cells. The data can be supported by the addition of a few simple controls, and a more detailed explanation of some of the methods, but generally speaking I am convinced by the conclusion that spermidine causes production of ROS by oxidizing Fe2+ in the cell. The protective role of other biomolecules such as RNA is more speculative, but nonetheless intriguing, and consistent with the genetic data. I feel that the *S. aureus* data add little to the paper. A big problem with the *S. aureus* experiments is that they compare two genetically distinct strains rather than comparing wild-type and mutant derivatives of otherwise genetically identical strains. Nonetheless, the *S. aureus* data are not needed, since the *E. coli* work stands on its own.

1. The conclusion from the data in Figure 1 seems to be that both high and low spermidine levels, relative to wild-type cells grown without exogenous spermidine, increase ROS levels. This is consistent with the overall conclusion that there is a "sweet spot" of spermidine levels needed to minimize ROS. This conclusion should be clearly stated in the Results to give context, especially since some of the data appear at first glance to be contradictory.

2. The data in Figure 2 involve comparisons of two strains of *S. aureus*, one of which has speG. The data are interpreted in a way that attributes all the phenotypic differences between the strains to the presence/absence of speG, but presumably there are many other genetic differences between these strains. The wording in this section of the paper should be softened to reflect the possibility that there could be other reasons for the phenotypic differences. Alternatively, the authors should make mutant strains that introduce or delete speG from the two strains, and test phenotypes in those genetic backgrounds.

3. Perhaps I missed it, but I couldn't see how the authors determined ("estimated", in their words) levels of NADP or glutathione, values for which are reported in Figure 3. The authors do mention a kit-based assay for measuring NAD levels, but there is insufficient description of this.

4. Line 195. There are lots of genes regulated by Fis or IHF. Is there a statistically significant enrichment of these genes among those differentially expressed, based on the microarray experiment?

5. Figure 4C. The authors gloss over the observation that SodA levels are similar in wt, untreated cells and spermidine-treated speG mutant cells. This is an unexpected result that warrants some discussion.

6. Figure 4 and associated text. This feels like a list of genes that went up or down, with little explanation of the expected result based on the prior observations in the paper. Hence, the significance of these observations is lost on the reader.

Figure 1C and 1E. The graphs would be easier to view if the lines were color-coded, especially for Figure 1E.

Line 143: "Therefore, spermidine stress would likely evoke O2- radicals in the *S. aureus* RN4220 but not in USA300 strain". This statement is too strong, since it is based on data from *E. coli*.

Line 163. "Confirms" is too strong a word to use here without measuring intracellular levels of the relevant molecules with/without each of the treatments. I recommend softening the wording to "is consistent with".

Figure 3 and associated description in the Results. The authors should make it clear at the start of this section and in the figure title that these experiments were done with *E. coli*.

Figure 4A. The authors should add a description of each of the numbered gene categories.

Figure 4C. The quantification graph is too small to easily interpret.

Figure 4G. The authors should plot data for Mn on a separate y-axis scale.

Figure 4G. This panel includes data for *E. coli* and data for *S. aureus*, which is very confusing. The authors should use a separate panel for data from each species.

Line 335. Should read "spermidine is highly toxic to cells".

*Reviewer #2:*

The authors have investigated why the polyamine spermidine is toxic to the bacterium *Escherichia coli* when intracellular spermidine is present in excess. To obtain cells where spermidine might be present in excess, they sought to create a gene deletion strain for the speG gene that encodes spermidine N-acetyltransferase, an enzyme that N-acetylates spermidine when spermidine levels are in excess, thereby neutralizing whatever function of spermidine is deleterious to cell growth. In principle, a speG gene deletion strain of *E. coli* is likely to have higher intracellular levels of spermidine. The authors also grew the speG strain in media contain high levels of added spermidine, which would be expected to further increase intracellular spermidine levels. A large number of physiological experiments were performed by the authors to demonstrate whether excess spermidine affects oxidative stress, and they concluded that it caused the generation of superoxide. Other analyses were performed to assess the affect of spermidine on iron oxidation. The global impact of spermidine toxicity was assessed by performing a transcriptional microarray experiment where the speG gene deletion strain was grown with and without 3.8 mM added spermidine in the growth medium, and the authors concluded that spermidine affects iron-sulfur cluster biogenesis.

Unfortunately (and inexplicably), the authors at no point measured spermidine or N-acetylspermidine levels in the cells that they were working with. The gene deletion regions for spermidine N-acetyltransferase (speG) and for spermidine synthase (speE) were transferred from KEIO collection strains into the authors' model *E. coli* wildtype strain by phage P1 transduction but were not confirmed by PCR after transduction, or by measuring spermidine or N-acetylspermidine levels. Thus, there is no proof that spermidine levels are altered in the speG or spermidine synthase (speE) gene deletion strains. The gene deletion strains were not complemented by the wildtype genes to show that any growth effects were not due to off-target damage cause by the P1 transduction. Furthermore, the supposed gene deletion strains were grown in LB medium, which contains spermidine, instead of in chemically defined M9 medium that does not contain spermidine. Because the spermidine content of the gene deletion strains is unknown, one can have very little confidence in any result that is supposedly based on whether cells contain excess spermidine or not. Although the microarray data looked at the effect of {plus minus} 3.8 mM spermidine on gene expression in the speG gene deletion strain, without knowledge of the intracellular spermidine content, any transcriptional effect could be due to indirect effects of spermidine on the cell wall, the outer membrane, on pH, and osmotic effects. In conclusion, the authors have performed a large number experiments where any conclusions on outcomes are in doubt due to the complete lack of knowledge of the content of intracellular spermidine, the metabolite that is supposedly responsible for the observed effects. Since, presumably, the authors do not have the data for intracellular spermidine content in their experiments, it is difficult to see how the data obtained can be used to make physiological conclusions, other than relating the observed effects to the presence or absence of added spermidine in the growth medium.

It is difficult to see how the data from this study can be salvaged. The observed effects cannot be attributed to excess spermidine or the complete absence of spermidine, since the level of spermidine in cells was never measured. Furthermore, the purported spermidine synthase (speE) deletion strain was grown in LB medium, which is a rich medium containing spermidine. Growth of the supposedly spermidine deficient speE deletion strain in LB medium will result in spermidine uptake from the LB medium, such that the supposedly depleted spermidine will be replaced by spermidine from the LB medium. The authors should have used chemically defined M9 medium for any spermidine-related experiment. Not only should spermidine levels and N-acetylspermidine levels have been measured in the speG deletion strain, but the levels of these metabolites should have been measured over the growth curve because N-acetylspermidine is accumulated more in stationary phase.

It may be possible to express the physiological changes observed in this study to the presence or absence of added spermidine to the growth medium, but the changes observed cannot be attributed to the levels of intracellular spermidine, since they were never measured.

*Reviewer #3:*

In the manuscript by Kumar et al., the authors proposed that sperimidine (SPD) triggers the production of superoxide radicals and claimed that this production is the unique molecular mechanism of toxicity in bacteria (line 81). These conclusions are based on the used of ROS probes. The results are interesting, provocative and conceptually new. However, clarifications, controls and new experiences are needed to support their conclusions.

My main concern is that the authors consider the ROS probes as specific for each compound (HO{degree sign}, O2{degree sign}-). The interconnection between ROS in the cells and the absence of great specificity of this type of tools should not make it possible to affirm the dependence of one type of ROS. Other approach would be necessary. Detecting very transient ROS, such as superoxide radicals, is challenging in the study of oxidative stress in vivo and an electron paramagnetic resonance (EPR) approach is considered ideal since it is unique for detecting free radicals.

The anaerobic experiments which support the superoxide conclusion need to be mitigated as polyamine uptake has been reported to be PMF dependant (Kashiwagi et al. J. Bact 1986). Therefore, a possible other explanation will be that in anaerobic condition PMF is slow down, polyamine uptake decrease and bacteria are more resistant. This point needs to be integrating in the text.

How the ROS probes response to SPD in anaerobic condition, what happens in the absence of oxygen if nitrate is present ?

The absence of SoxR activation in presence of spermidine is difficult to understand. The authors could used the Imlay study (Gu et al. 2011 , DOI: 10.1111/j.1365-2958.2010.07520.x) in which it is show that SoxR is directly activated by redox cycling drugs rather than by superoxide.

Figure 1 A and B : add control (wt+SPD ; wt+SPD +/- tiron).

Figure 1C : speE mutant is expected to have less SPD and therefore more H2O2, how the authors explain the opposite result?

Figure 1E : add control (WT+pSodA ; speG+pSodA) and all the strains with empty vector

Sup1 : C : does the sod(A or B) simple mutant are more sensitive to SPD at higher concentration than 4.5mM?

Figure 3 B and C : add control (empty vector).

Figure 4A : microarray result, the authors could add and comment the data on the suf operon (Fe/S cluster biosynthesis during stress condition, ROS and iron limitation).

[Editors’ note: further revisions were suggested prior to acceptance, as described below.]

Thank you for resubmitting your work entitled "Free spermidine evokes superoxide radicals that manifest toxicity" for further consideration by eLife. Your revised article has been evaluated by Gisela Storz (Senior Editor) and a Reviewing Editor.

The manuscript has been improved, and the reviewers are satisfied that you have addressed the major concerns from the previous round of review. Nevertheless, there are a few small issues that still need to be addressed, as detailed in the two reviews below:

*Reviewer #1:*

The authors have responded positively to the reviewers' comments. I have only a few comments:

1. Figure 1A. MFI levels for H2DCFDA are reduced when spermidine is added to wild-type cells, but the intracellular concentration of spermidine does not detectably increase in these cells (Figure 1 - Figure Supplement 1C). Thus, the data don't support the conclusion that ROS detected by H2DCFDA is reduced as a result of increased intracellular spermidine levels.

2. Line 124-5, "...although spermidine accumulation in the ΔspeG strain reduces overall ROS levels and oxidative stress...". This statement is misleading. Intracellular levels of spermidine are higher in the ΔspeG strain when spermidine is added exogenously, and these cells do exhibit reduced H2DCFDA fluorescence, but so do wild-type cells, where the intracellular spermidine levels do not increase when spermidine is added exogenously.

3. The authors should show their HPLC traces used for quantifying intracellular spermidine levels. These data should include the control data using purified spermidine, and should indicate the quantified area of the HPLC trace.

4. Lines 140-1, "The double and triple mutants containing empty vector exhibited higher growth defects than ΔspeG strain on LB-agar plate supplemented with spermidine (Figure 1E)." I am not convinced by this conclusion; the sodA deletion has a very small effect when combined with the speG deletion, and the sodB deletion has no effect. However, what is clear is that overexpression of sodA rescues the growth defect of a speG mutant grown with exogenous spermidine. The authors don't mention this important result.

5. Figure 4 - Figure Supplement 1. The authors should repeat this restreak so that individual colonies for each strain can be more easily compared.

6. It seems likely that Fis-regulated and IHF-regulated genes are enriched in the set of genes identified as being differentially expressed in the microarray analysis. However, the authors have not done a statistical test to address this. A Fisher's exact test would suffice for this.

*Reviewer #2:*

I'm very happy to see that some of my suggestions from the first review were implemented by the authors.

Attached are a few more comments.

1) Claiming that the production of superoxyde is the unique molecular mechanism of toxicity is always too strong.

lines 35-36: "Therefore, we propose that the spermidine-induced superoxide radicals cause spermidine toxicity in *E. coli*"

line 84: "we decipher a unique molecular mechanism of spermidine toxicity in bacteria"

whereas in the rebuttal letter, the authors are more cautious, which is more appropriate: "superoxide generation is one of the major causes of spermidine toxicity" (just before reviewer#2 recommendations for the authors).

2) The comments on *S. aureus* in the introduction (l.73-77) are not any more necessary

3) Fig.1E: all the strains carrying plasmids (pDAK1 or pSodA), the plasmids are missing for ∆speG ∆sodB and ∆speG ∆sodA ∆sodB: is it a mistake? I hope so because otherwise, it lacks this control. Moreover, it is not indicated the presence of antibiotic to maintain plasmids in the agar plate +/- SPD (methods line 448 and legend).

4) lines 140, 146 and 297: the use of single, double or triple mutants is not clear, please rephrase

5) lines 168 to 176: errors on the panels of EPR experiments in the text:

(2B and 2C) instead of (2B and 2E)

(2D and 2E) instead of (2C and 2F)

(2F and 2G) instead of (2D and 2G)

6) Line 362 : "This is why spermidine is a double-edged sword where in excess, it provokes O2- anion production, and in scarcity, it leads to general ROS production."

Lower SPD level does not lead to ROS production but to a higher ROS level, please rephrase.

---

## [Author Response]

[Editors’ note: the authors resubmitted a revised version of the paper for consideration. What follows is the authors’ response to the first round of review.]

The reviewers raised four major concerns. First, the probes used to determine the type of ROS stress are not specific enough to confidently draw the conclusion that spermidine causes O2- production, as opposed to another ROS. Hence, we would want to see a more direct test of the specific ROS produced. Electron paramagnetic resonance would be a suitable approach for this experiment. Second, several of the experiments use deletion mutant strains but lack a control where the deletion has been complemented, together with a corresponding empty-vector control. Third, the reviewers felt that the S. aureus experiments are peripheral to the main theme of the paper, and should be removed. Fourth, the reviewers were concerned that intracellular spermidine levels for different strains/growth conditions were inferred but not directly measured, particularly for the experiments in Figure 1 with speE and speG mutants and with spermidine supplementation. The use of LB as a growth medium is a concern in this regard, since LB contains spermidine. We would want to see either measurement of intracellular spermidine levels for wild-type and mutant strains grown under the conditions used in the corresponding experiments, or references to prior work where spermidine levels have been measured under the same growth conditions for the same strains. We believe these concerns are addressable, but would require a considerable amount of work, which would take more than two months. Hence, we are rejecting the paper, but we would consider a revised version if you are able to address the four major concerns.

We addressed all four issues, listed above, before resubmitting the manuscript:

1. EPR experiments were performed to confirm spermidine-mediated O2-production in ∆*speG* cells.

2. Now we have shown phenotypes with deletion and complementation along with proper empty vector controls.

3. *S. aureus* data has been removed.

4. HPLC-coupled MS analyses was done to show the levels of spermidine in different *E. coli* cells.

Lastly, the editor was rightly pointed out that the addressing the above concerns would take more than two months. Due to COVID situation and other inconveniences, we had to take four months to address the points carefully.

The response to the reviewer’s specific comments are as follows:

Reviewer #1:The work presented in this study provides important insight into the role of spermidine in bacterial cells. The data can be supported by the addition of a few simple controls, and a more detailed explanation of some of the methods, but generally speaking I am convinced by the conclusion that spermidine causes production of ROS by oxidizing Fe2+ in the cell. The protective role of other biomolecules such as RNA is more speculative, but nonetheless intriguing, and consistent with the genetic data. I feel that the S. aureus data add little to the paper. A big problem with the S. aureus experiments is that they compare two genetically distinct strains rather than comparing wild-type and mutant derivatives of otherwise genetically identical strains. Nonetheless, the S. aureus data are not needed, since the E. coli work stands on its own.

*S. aureus* data is removed, as suggested.

1. The conclusion from the data in Figure 1 seems to be that both high and low spermidine levels, relative to wild-type cells grown without exogenous spermidine, increase ROS levels. This is consistent with the overall conclusion that there is a "sweet spot" of spermidine levels needed to minimize ROS. This conclusion should be clearly stated in the Results to give context, especially since some of the data appear at first glance to be contradictory.

Now we have rewritten our results to mention the point clearly. See line numbers 102-104, 116-119, 122-127 in the main text.

2. The data in Figure 2 involve comparisons of two strains of S. aureus, one of which has speG. The data are interpreted in a way that attributes all the phenotypic differences between the strains to the presence/absence of speG, but presumably there are many other genetic differences between these strains. The wording in this section of the paper should be softened to reflect the possibility that there could be other reasons for the phenotypic differences. Alternatively, the authors should make mutant strains that introduce or delete speG from the two strains, and test phenotypes in those genetic backgrounds.

*S. aureus* data is removed, as suggested.

3. Perhaps I missed it, but I couldn't see how the authors determined ("estimated", in their words) levels of NADP or glutathione, values for which are reported in Figure 3. The authors do mention a kit-based assay for measuring NAD levels, but there is insufficient description of this.

The measurements are not absolute levels but a relative analysis. We removed the word estimated “and coined “compared the levels” in the text (Line number 193). We elaborated it in the methodology section (Lines 577-607)

4. Line 195. There are lots of genes regulated by Fis or IHF. Is there a statistically significant enrichment of these genes among those differentially expressed, based on the microarray experiment?

Thanks to the reviewer for this comment. As per the current EcoCyc entry, there are 137 and 106 transcriptional units in *E. coli* that are regulated by Fis and IHF, respectively. In our microarray, we find 45 Fis-regulated genes, 13 IHF regulated genes, and 4 Fis+IHF regulated genes have altered expression. We think that these numbers are significant especially the numbers of the Fis-regulated genes. Accordingly, ∆*speG*∆*fis* mutant, but not the ∆*speG*∆*ihfA* mutant, grows slowly than the ∆*speG* mutant. We just mention this number of genes in the text (Line 219).

5. Figure 4C. The authors gloss over the observation that SodA levels are similar in wt, untreated cells and spermidine-treated speG mutant cells. This is an unexpected result that warrants some discussion.

I think reviewer missed the texts where we expressed our concerns for this unexpected result saying spermidine may directly or indirectly inhibit superoxide-mediated induction of SoxR in the old text. However, further we modified our text in the light of two alternative explanations: Late Prof. Fridovich and many initial work had opined that superoxide can directly activate SoxR. On the other hand, Prof. Imlay group has shown that SoxR is efficiently activated by redox cycling drugs, but not by superoxide. We observed superoxide generation and no activation of SoxR which fits well with the second school of thought. However, we have also used a redox cycling drug, menadione, which activated SoxR greatly, and spermidine deactivates this menadione-induced SoxR activation. This data is pointing towards a bigger and generalized concept that spermidine may keep SoxR in apo-SoxR form by interfering its iron-sulfur cluster biogenesis. As a result, be it superoxide or redox-cycling drug, they cannot activate SoxR! We mentioned these new points in different part of the revised manuscript (Lines: 235-241, 431-438).

6. Figure 4 and associated text. This feels like a list of genes that went up or down, with little explanation of the expected result based on the prior observations in the paper. Hence, the significance of these observations is lost on the reader.

Added lines 215-218 in the revised main text.

Figure 1C and 1E. The graphs would be easier to view if the lines were color-coded, especially for Figure 1E.

Now revised panels were color-coded in Figure 1.

Line 143: "Therefore, spermidine stress would likely evoke O2- radicals in the S. aureus RN4220 but not in USA300 strain". This statement is too strong, since it is based on data from E. coli.

*S. aureus* data has been removed as recommended by Editor and reviewer I.

Line 163. "Confirms" is too strong a word to use here without measuring intracellular levels of the relevant molecules with/without each of the treatments. I recommend softening the wording to "is consistent with".

Replaced accordingly.

Figure 3 and associated description in the Results. The authors should make it clear at the start of this section and in the figure title that these experiments were done with E. coli.

Since *S. aureus* data has been removed, this will not be a problem now.

Figure 4A. The authors should add a description of each of the numbered gene categories.

The numbered gene categories are listed in the supplementary information section. Now, we mentioned the same thing in the Figure 4A legend.

Figure 4C. The quantification graph is too small to easily interpret.

We redistributed the panels of Figure 4 into two different figures (Figure 4 and a new Figure 5) to show the quantification graph prominently.

Figure 4G. The authors should plot data for Mn on a separate y-axis scale.

Done. Instead of putting a separate Y axis, two different plots were made; one for Fe (Figure 4B) and another for Mn (Figure 4D).

Figure 4G. This panel includes data for E. coli and data for S. aureus, which is very confusing. The authors should use a separate panel for data from each species.

*S. aureus* data is removed as recommended by the reviewer and Editor.

Line 335. Should read "spermidine is highly toxic to cells".

Removed the article “The” from the sentence.

Reviewer #2:The authors have investigated why the polyamine spermidine is toxic to the bacterium Escherichia coli when intracellular spermidine is present in excess. To obtain cells where spermidine might be present in excess, they sought to create a gene deletion strain for the speG gene that encodes spermidine N-acetyltransferase, an enzyme that N-acetylates spermidine when spermidine levels are in excess, thereby neutralizing whatever function of spermidine is deleterious to cell growth. In principle, a speG gene deletion strain of E. coli is likely to have higher intracellular levels of spermidine. The authors also grew the speG strain in media contain high levels of added spermidine, which would be expected to further increase intracellular spermidine levels. A large number of physiological experiments were performed by the authors to demonstrate whether excess spermidine affects oxidative stress, and they concluded that it caused the generation of superoxide. Other analyses were performed to assess the affect of spermidine on iron oxidation. The global impact of spermidine toxicity was assessed by performing a transcriptional microarray experiment where the speG gene deletion strain was grown with and without 3.8 mM added spermidine in the growth medium, and the authors concluded that spermidine affects iron-sulfur cluster biogenesis.Unfortunately (and inexplicably), the authors at no point measured spermidine or N-acetylspermidine levels in the cells that they were working with. The gene deletion regions for spermidine N-acetyltransferase (speG) and for spermidine synthase (speE) were transferred from KEIO collection strains into the authors' model E. coli wildtype strain by phage P1 transduction but were not confirmed by PCR after transduction, or by measuring spermidine or N-acetylspermidine levels. Thus, there is no proof that spermidine levels are altered in the speG or spermidine synthase (speE) gene deletion strains. The gene deletion strains were not complemented by the wildtype genes to show that any growth effects were not due to off-target damage cause by the P1 transduction. Furthermore, the supposed gene deletion strains were grown in LB medium, which contains spermidine, instead of in chemically defined M9 medium that does not contain spermidine. Because the spermidine content of the gene deletion strains is unknown, one can have very little confidence in any result that is supposedly based on whether cells contain excess spermidine or not. Although the microarray data looked at the effect of {plus minus} 3.8 mM spermidine on gene expression in the speG gene deletion strain, without knowledge of the intracellular spermidine content, any transcriptional effect could be due to indirect effects of spermidine on the cell wall, the outer membrane, on pH, and osmotic effects. In conclusion, the authors have performed a large number experiments where any conclusions on outcomes are in doubt due to the complete lack of knowledge of the content of intracellular spermidine, the metabolite that is supposedly responsible for the observed effects. Since, presumably, the authors do not have the data for intracellular spermidine content in their experiments, it is difficult to see how the data obtained can be used to make physiological conclusions, other than relating the observed effects to the presence or absence of added spermidine in the growth medium.

I appreciate the concern of the Reviewer. In fact, many years ago, we have noticed that some knockouts in Keio collection (e.g ∆*hns*) are not perfect. Therefore, our default lab practice is to make fresh strains by P1 transduction in an isogenic WT *E. coli* background (Either MG1655 or BW25113) followed by confirming the allelic replacement by PCR using flanking primer-pairs and Sanger sequencing. Although we had already mentioned the point of PCR verification in the Methods section, we forgot to mention the Sanger sequencing part. Now we clearly mentioned the sanger sequencing part (Line number: 416-420). in Methods section.

Now we have performed HPLC coupled MS analyses of the tri-dansylated-spermidine to determine its intracellular level to support our observations (in collaboration with my colleague Dr. Vinod D. Chaudhari; See the new author list for the detail). Di-dansylated-hexane diamine was used as an internal standard. Polyamines were extracted from about 50mg wet cell pellets adding internal standard and then fully dansylated. The tri-dansylated spermidine peak areas were first normalized with the area of the internal standards, and then with the wet mass of the cell pellets. A standard curve was also generated using different amounts of tri-dansylated spermidine using the HPLC. The cellular levels of spermidine were represented as µmol/100mg of wet cell pellets (Figure 1—figure supplement 1C and Source data Figure 1—figure supplement 1C).

We could not resolve N-acetyl spermidine peak using the protocol used by Dr. Chaudhari lab. We see an old literature where authors have shown two different, N1 and N8-acetyl spermidine peaks. In our case these two acetylated peaks but were merged with many other components, as detected in MS data.

We agree with the reviewers that the spermidine can exerts effect on cell wall, outer membrane, pH and osmotic balance but how it will cause different effects to WT and ∆*speG* cells when present in extracellular environment is not conceivable. They can cause differential effect on those components only when accumulate intracellularly which was possible in ∆*speG* cells, but not in a wild type background. For example, spermidine also has an effect on translation. However, we have shown here that the superoxide generation is one of the major cause of spermidine toxicity. Thus, effects of spermidine in two isogenic strains (WT and DspeG) are different.

Our data now shows the intracellular level of spermidines in the different strains (Figure 1—figure supplement 1C and Source data Figure 1—figure supplement 1C).

It is difficult to see how the data from this study can be salvaged. The observed effects cannot be attributed to excess spermidine or the complete absence of spermidine, since the level of spermidine in cells was never measured. Furthermore, the purported spermidine synthase (speE) deletion strain was grown in LB medium, which is a rich medium containing spermidine. Growth of the supposedly spermidine deficient speE deletion strain in LB medium will result in spermidine uptake from the LB medium, such that the supposedly depleted spermidine will be replaced by spermidine from the LB medium. The authors should have used chemically defined M9 medium for any spermidine-related experiment. Not only should spermidine levels and N-acetylspermidine levels have been measured in the speG deletion strain, but the levels of these metabolites should have been measured over the growth curve because N-acetylspermidine is accumulated more in stationary phase.It may be possible to express the physiological changes observed in this study to the presence or absence of added spermidine to the growth medium, but the changes observed cannot be attributed to the levels of intracellular spermidine, since they were never measured.

Now we have performed MS-coupled HPLC analyses of the spermidine peaks to determine its intracellular level to support our observations.

To get an idea, we compared the relative spermidine peak area in the LB medium to WT cell. This suggests that dry LB medium contains approx. 25 times lesser spermidine concentration than equal wet mass of WT cell pellets. If we consider the bio-number, The 100 mg wet cell pellet will correspond to about 20 mg dry mass. Therefore, comparing dry masses, LB powder will have 125 time lesser concentration of spermidine than in *E. coli* Dry cell pellets. Thus, this spermidine contamination in LB medium transported uphill to raise the polyamine level in the ∆*speE* strain to some extent, but it remains well below WT levels.

We could not resolve N-acetyl spermidine peaks but could detect their presence or absence in the MS data.

Reviewer #3:In the manuscript by Kumar et al., the authors proposed that sperimidine (SPD) triggers the production of superoxide radicals and claimed that this production is the unique molecular mechanism of toxicity in bacteria (line 81). These conclusions are based on the use of ROS probes. The results are interesting, provocative and conceptually new. However, clarifications, controls and new experiences are needed to support their conclusions.

Now we have come up with new experiments and clarifications as stated point-by point below.

My main concern is that the authors consider the ROS probes as specific for each compound (HO{degree sign}, O2{degree sign}-). The interconnection between ROS in the cells and the absence of great specificity of this type of tools should not make it possible to affirm the dependence of one type of ROS. Other approach would be necessary. Detecting very transient ROS, such as superoxide radicals, is challenging in the study of oxidative stress in vivo and an electron paramagnetic resonance (EPR) approach is considered ideal since it is unique for detecting free radicals.

Thanks to the reviewer for educating us with his helpful comments. We are convinced by the reviewer that the ROS probes, specifically H2DCFDA or DHE are not very specific for OH or superoxide radicals! Although DHE would be somewhat specific for superoxide in our assay condition. However, we cannot take any chance to misinterpret the observations. Therefore, as per the reviewer’s suggestion, we performed EPR spectroscopy experiments for superoxide detection.

We have struggled a lot to execute the suggested EPR experiments for superoxide species detection. This is because the EPR machines are available in a few distantly places in India and either heavily occupied by the physicists and chemists, or usage were restricted due to COVID situation. We just could just manage to get few slots in two different places, to perform the bare minimum experiments to standardize and get the results (with ∆*speG* strain plus minus spermidine) to prove our points.

We used CMH spin-probe for superoxide detection as it is cell permeable, required to add in low concentration, reacts better with superoxide and half life is greater than the spin traps and other spin probes. Quickly, just raw panels that are shown in Author response image 1, were used to make the new Figure 2 (replacing old figure 2 of *S. aureus* data) in the manuscript. Also, we added a section in the result and another in the Materials and methods (Line numbers 148-175; 473-485).

**Author response image 1. sa2fig1:** 

The anaerobic experiments which support the superoxide conclusion need to be mitigated as polyamine uptake has been reported to be PMF dependant (Kashiwagi et al. J. Bact 1986). Therefore, a possible other explanation will be that in anaerobic condition PMF is slow down, polyamine uptake decrease and bacteria are more resistant. This point needs to be integrating in the text.

Now, we have integrated this point in the text as recommended by the reviewer. Also added this reference:

Kashiwagi K, Kobayashi H, Igarashi K. Apparently unidirectional polyamine transport by proton motive force in polyamine-deficient *Escherichia coli*. J Bacteriol. 1986 Mar;165(3):972-7. doi: 10.1128/jb.165.3.972-977.1986. PMID: 3005244; PMCID: PMC214524.

How the ROS probes response to SPD in anaerobic condition, what happens in the absence of oxygen if nitrate is present?

They did not differ in signal between the conditions.

The absence of SoxR activation in presence of spermidine is difficult to understand. The authors could used the Imlay study (Gu et al. 2011 , DOI: 10.1111/j.1365-2958.2010.07520.x) in which it is show that SoxR is directly activated by redox cycling drugs rather than by superoxide.

Now, we have included this possibility in the results, and discussion that superoxide may not directly activate the SoxR (Lines: 235-241, 431-438).

Figure 1 A and B : add control (wt+SPD ; wt+SPD +/- tiron).

Controls added.

Figure 1C : speE mutant is expected to have less SPD and therefore more H2O2, how the authors explain the opposite result?

Thanks for pointing to this this crucial point.

We repeated the experiments and again find that the H2DCFDA and DHE signals were higher (see new Figure 1A and B), but H2O2 levels was again lower in ∆*speE* strain than WT cells under spermidine stress. We also checked KatG and AhpC levels. They do not change in ∆*speE* strain under spermidine stress! Therefore, indeed these paradoxical observations are difficult to explain. One possibility could be that in the absence of spermidine, the level of free irons are more (as we have shown spermidine interacts with free iron) that may participate in Fenton reaction consuming more H2O2 and liberating Hydroxyl and hydroperoxyl radicals (see the reactions in Author response image 2). However, this is just a hypothesis.

Figure 1E : add control (WT+pSodA ; speG+pSodA) and all the strains with empty vector.

Now we have added controls as suggested.

Sup1 : C : does the sod(A or B) simple mutant are more sensitive to SPD at higher concentration than 4.5mM?

We did not find any difference (see Author response image 3). This result could be due to non-induction of sodA/B, and/or declined level of manganese. Further, we do not know at this moment if functional gain of SpeG in the *sod* mutant strains, if any. The part of the Sup1C figure (SPD 4.5mM) has now been moved to main Figure 1. Rest of the Sup1C has been moved to Sup 4.

**Author response image 3. sa2fig3:** 

Figure 3 B and C : add control (empty vector).

Added and repeated the experiments

Figure 4A : microarray result, the authors could add and comment the data on the suf operon (Fe/S cluster biosynthesis during stress condition, ROS and iron limitation).

Thanks for this suggestion. We have seen *suf* operon was indifferent except *sufA* was about 2.5-fold downregulated. The *sufA* expression data is now added in heat-map (Figure 4a). We explained these results at related texts (270-277). Also, added the sufA expression in microarray heat-map panel (Figure 4A)

[Editors’ note: what follows is the authors’ response to the second round of review.]

Reviewer #1:The authors have responded positively to the reviewers' comments. I have only a few comments:1. Figure 1A. MFI levels for H2DCFDA are reduced when spermidine is added to wild-type cells, but the intracellular concentration of spermidine does not detectably increase in these cells (Figure 1 - Figure Supplement 1C). Thus, the data don't support the conclusion that ROS detected by H2DCFDA is reduced as a result of increased intracellular spermidine levels.

Indeed, in the light of the above data, it will be erroneous to say that increased spermidine level in WT cells by exogenous spermidine stress results in decreased H2DCFDA fluorescence. However, one cannot deny that exogenous spermidine supplementation caused the decreased H2DCFDA fluorescence. In fact, SpeG (spermidine N-acetyl transferase) enzyme can quickly acetylate either N1 and N8-amine groups of excess spermidines forming two different species of N-acetyl-spermidines in WT strain keeping almost constant level of spermidine. In eukaryotic system, there is known mechanisms of further degradation/recycling of N-acetyl-spermidines, but nothing is known about the fate of N-acetyl spermidines in prokaryotes. Furthermore, whether N-acetyl spermidines, the products of spermidine overaccumulation in WT cells, have also roles in decreasing ROS is not known so far.

On the other hand, Δ*speG* strain is devoid of SpeG function, and therefore, there would be very little (if spontaneous acetylation exists), or no turnover of spermidine to N-acetyl-spermidines. Therefore, one can easily detect spermidine overaccumulation.

Therefore, we never claimed (in line number 123-125) that increased intracellular spermidine level in WT cells caused the decreased H2DCFDA fluorescence.

To further clarify the issue, we have added a few extra texts (Lines: 95-97; 112; 114-118) in the manuscript.

2. Line 124-5, "...although spermidine accumulation in the ΔspeG strain reduces overall ROS levels and oxidative stress...". This statement is misleading. Intracellular levels of spermidine are higher in the ΔspeG strain when spermidine is added exogenously, and these cells do exhibit reduced H2DCFDA fluorescence, but so do wild-type cells, where the intracellular spermidine levels do not increase when spermidine is added exogenously.

Already clarified this issue in the above-mentioned response against reviewer’s recommendation point no 1 (Lines: 95-97; 112; 114-118).

3. The authors should show their HPLC traces used for quantifying intracellular spermidine levels. These data should include the control data using purified spermidine, and should indicate the quantified area of the HPLC trace.

We have added this HPLC profile pictures in a separate source data (Figure 1 – figure supplement 1C - Source data 2)

4. Lines 140-1, "The double and triple mutants containing empty vector exhibited higher growth defects than ΔspeG strain on LB-agar plate supplemented with spermidine (Figure 1E)." I am not convinced by this conclusion; the sodA deletion has a very small effect when combined with the speG deletion, and the sodB deletion has no effect. However, what is clear is that overexpression of sodA rescues the growth defect of a speG mutant grown with exogenous spermidine. The authors don't mention this important result.

We have modified the text mentioning the strain names (Lines 142-144). Also, we have mentioned that the sodA overexpression rescued the growth defect of speG mutant (Lines 146-149)

5. Figure 4 - Figure Supplement 1. The authors should repeat this restreak so that individual colonies for each strain can be more easily compared.

Repeated and the old figure has been replaced.

6. It seems likely that Fis-regulated and IHF-regulated genes are enriched in the set of genes identified as being differentially expressed in the microarray analysis. However, the authors have not done a statistical test to address this. A Fisher's exact test would suffice for this.

First, we distributed 49 Fis-enriched genes in 32 transcriptional units/operons as per EcoCyc database. Similarly, 17 IHF-enriched genes were distributed in 13 transcriptional units/operons. We also checked the total number of transcriptional units/operons that are upregulated or downregulated by the above two transcription regulators. Then Fisher’s exact test for Fis-enriched and IHF-enriched transcriptional units were performed. The calculated P-values for the Fis-enriched and IHF-enriched transcriptional units were 0.0023 (i.e., significant) and 0.7714 (i.e., not significant), respectively. We have mentioned this Fisher’s exact test in the result section (Line number 224-225) to draw its significance.

Reviewer #2:I'm very happy to see that some of my suggestions from the first review were implemented by the authors.Attached are a few more comments.1) Claiming that the production of superoxyde is the unique molecular mechanism of toxicity is always too strong.lines 35-36: "Therefore, we propose that the spermidine-induced superoxide radicals cause spermidine toxicity in E. coli"line 84: "we decipher a unique molecular mechanism of spermidine toxicity in bacteria"whereas in the rebuttal letter, the authors are more cautious, which is more appropriate: "superoxide generation is one of the major causes of spermidine toxicity" (just before reviewer#2 recommendations for the authors).

We have made the necessary changes in those lines, as recommended.

2) The comments on S. aureus in the introduction (l.73-77) are not any more necessary

We have now deleted this text from the revised manuscript.

3) Fig.1E: all the strains carrying plasmids (pDAK1 or pSodA), the plasmids are missing for ∆speG ∆sodB and ∆speG ∆sodA ∆sodB: is it a mistake? I hope so because otherwise, it lacks this control. Moreover, it is not indicated the presence of antibiotic to maintain plasmids in the agar plate +/- SPD (methods line 448 and legend).

We are extremely sorry for this inadvertent mistake! Now we rectified it.

4) lines 140, 146 and 297: the use of single, double or triple mutants is not clear, please rephrase

We modified the lines by mentioning the exact names of those mutants (new Lines: 140-143; 298-300).

5) lines 168 to 176: errors on the panels of EPR experiments in the text:(2B and 2C) instead of (2B and 2E)(2D and 2E) instead of (2C and 2F)(2F and 2G) instead of (2D and 2G)

Thank you for pointing out this error! We changed them accordingly (new lines 171-180).

6) Line 362 : "This is why spermidine is a double-edged sword where in excess, it provokes O2- anion production, and in scarcity, it leads to general ROS production."Lower SPD level does not lead to ROS production but to a higher ROS level, please rephrase.

We rephrased the text accordingly (new Lines 361-362)